# From creep to flow: Granular materials under cyclic shear

Ye Yuan [1], Zhikun Zeng [1], Yi Xing[1], Houfei Yuan[1], Shuyang Zhang[1], Walter Kob [2,3] & Yujie Wang [1,2,4]

When unperturbed, granular materials form stable structures that resemble the ones of other amorphous solids like metallic or colloidal glasses. Whether or not granular materials under shear have an elastic response is not known, and also the influence of particle surface roughness on the yielding transition has so far remained elusive. Here we use X-ray tomography to determine the three-dimensional microscopic dynamics of two granular systems that have different roughness and that are driven by cyclic shear. Both systems, and for all shear amplitudes $\Gamma$ considered, show a cross-over from creep to diffusive dynamics, indicating that rough granular materials have no elastic response and always yield, in stark contrast to simple glasses. For the system with small roughness, we observe a clear dynamic change at $\Gamma \approx 0.1$, accompanied by a pronounced slowing down and dynamical heterogeneity. For the large roughness system, the dynamics evolves instead continuously as a function of $\Gamma$. We rationalize this roughness dependence using the potential energy landscape of the systems: The roughness induces to this landscape a micro-corrugation with a new length scale, whose ratio over the particle size is the relevant parameter. Our results reveal the unexpected richness in relaxation mechanisms for real granular materials.

Yielding of amorphous materials is ubiquitous[1,2], governing a wide range of phenomena like mechanical failure of metallic glasses[3], complex rheologies of soft glasses[4,5], or geophysical catastrophes[6,7]. A typical amorphous solid under quasistatic simple shear will deform elastically at a small strain and flow plastically beyond the yielding strain. Depending on whether the material is brittle or ductile, the stress-strain curve shows a drop or a smooth cross-over to a plateau, respectively. On the particle level, yielding is a cooperative phenomenon of local plastic events[8,9], and computer simulations hint that this phenomenon shares many aspects with a first-order phase transition[10–12]. However, for real systems, the precise nature of this transition has not yet been clarified since such experiments are very challenging.

In contrast to standard glasses, for granular systems, there is at present no satisfactory understanding of yielding on the level of the particles. Theoretical arguments suggest that sheared granular solids are marginally stable, i.e., contacts between particles will change irreversibly even under a tiny applied strain, which implies that such systems have no elastic regime[13]. This view is at odds with experimental results which found that the stress-strain curve of granular solids under simple shear does in fact resemble the one of a glass[14]. One possibility to resolve this discrepancy is to consider a different type of driving, i.e., cyclic shear, since it allows to probe directly the reversibility of the particle trajectories and hence the presence of elasticity in the system. A number of computational studies have in fact used this setup to investigate the particle dynamics in soft-sphere jammed states and model glasses[15–26]. These works, as well as related experiments on soft glasses[27–30], have indeed revealed a transition at which the motion changes from reversible (or caged) to diffusive when the

[1]School of Physics and Astronomy, Shanghai Jiao Tong University, Shanghai 200240, China. [2]Department of Physics, College of Mathematics and Physics, Chengdu University of Technology, Chengdu 610059, China. [3]Department of Physics, University of Montpellier and CNRS, 34095 Montpellier, France. [4]State Key Laboratory of Geohazard Prevention and Geoenvironment Protection, Chengdu University of Technology, Chengdu 610059, China.
e-mail: walter.kob@umontpellier.fr; yujiewang@sjtu.edu.cn

cyclic shear amplitude $\Gamma$ is increased, thus supporting the existence of an elastic behavior if $\Gamma \lesssim 0.1$. However, these findings are in contrast with several experimental results for cyclically sheared granular materials, which display robust sub-diffusive dynamics instead of a caged regime for the values of $\Gamma$ considered, i.e., $0.07 \leq \Gamma \leq 0.26$[31–33]. Reversible trajectories are only found for sufficiently small $\Gamma \ll 0.1$[34]. The inconsistency between these studies as well as the complex dependence on the shear protocol[34,35] have not been elucidated.

We note that only few of the related simulations have included friction[16,35], an essential component of granular materials. Also, the connection between surface roughness and friction of granular particles has been investigated only recently[36–38], but their influence on the particle-level relaxation dynamics remains unknown, especially considering the experimental challenges in a three-dimensional granular system[39–41]. Advancing on these points is important not only for understanding the plasticity of amorphous materials but also for developing reliable granular constitution laws[42].

In the present work, we experimentally probe the microscopic dynamics of cyclically sheared granular materials in three dimensions. The two granular systems, with different particle surface roughness, show for all $\Gamma$ a crossover from creep (subdiffusive) to diffusive dynamics, in stark contrast to simple glasses. A dynamic change at $\Gamma \approx 0.1$ is clearly observed for the system with small roughness, but not for the one with large roughness. This suggests that roughness induces a micro-corrugation to the potential energy landscape that allows the activation of a novel relaxation mechanisms for granular materials.

## Results

### Cyclic shear experiment

We investigate experimentally a cyclically sheared granular system using an X-ray tomography technique[32,33,43,44] (see Methods for details). As shown in the schematic of Fig. 1a, particles are placed in a shear cell and cyclically sheared with an amplitude $\Gamma$[32,45]. This design allows to accommodate the granular compaction/dilation during shear. We utilize two types of spherical beads to check the influence of surface roughness: One is acrylonitrile butadiene styrene plastic (ABS) and the other is 3D-printed (ProJet MJP 2500 Plus, 0.032 mm resolution) with a bumpy surface (BUMP), see Fig. 1b. For both cases, the system contains $\approx 14{,}000$ 50:50 bidisperse beads of diameters 5 mm and 6 mm, and no crystallization was detected. The size of the shear cell at shear $\gamma = 0$ is $24d \times 24d \times 24d$, where $d$ is the diameter of the small beads, and which will be taken as the unit length. The strain rate is small so we are in quasistatic conditions with an inertia number less than $10^{-3}$ [46].

Beads are initially placed in the cell, forming reproducible loose packings, and then compacted by cyclic shear until the steady state is reached (see Supplementary Fig. 1). Thus none of the presented results are affected by transient effects. Since the beads are very rigid, the obtained packings always maintain mechanical stability under gravity and slow external driving. X-ray tomography scans are taken at $\gamma = 0$ (periodicity between 1 and 10 cycles) from which we extract the microscopic structure of the system and the dynamics of the particles. The resolution in particle position is around $10^{-3}d$. For the ABS system the reported results have been obtained by averaging for each $\Gamma$ over $3 - 5$ independent realizations, while only one realization is used for the BUMP system. To mitigate finite size effects we exclude for the analyses the particles located closer than $4d$ from the boundaries.

### Steady-state dynamics

The dynamics of the particles are basically isotropic, although convection (see Supplementary Fig. 2) and height dependence (see Supplementary Fig. 3) emerge very mildly due to gravity for large $\Gamma$. Hence we focus here on the dynamics in the (horizontal) $x$-and $y$-directions. Figure 2a and b present, respectively, the mean squared displacement (MSD), $\langle \delta x^2 (\Delta n) \rangle$, for the ABS and BUMP particles as a function of the number of cycles $\Delta n$, where $\delta x$ is the average displacement in the $x$-or

$y$-direction, and $\langle . \rangle$ denotes the average over different particles, starting configurations, and realizations. For both systems and all $\Gamma$, we find diffusive growth at large $\Delta n$ and subdiffusion at small $\Delta n$. Note that for each system the subdiffusion exponent is independent of $\Gamma$, i.e., 0.65 for ABS and 0.8 for BUMP. Such a universal creep dynamics at small $\Delta n$ demonstrates that our system has no caging or elastic regime, in agreement with earlier findings[31–33]. This is also confirmed by the absence of a two-step relaxation in the self-intermediate scattering function (see Supplementary Fig. 4).

In the following, we report the dynamics as a function of the accumulated strain, i.e., $\Delta \gamma = 4\Delta n \Gamma$, which takes into account one part of the expected $\Gamma$-dependence of the dynamics. Figure 2c shows that the $\Gamma$-dependence of the diffusion coefficient $D$, obtained from the Einstein relation $\langle \delta x^2 \rangle = 2D\Delta \gamma$, depends on the system considered: For the ABS system, $D$ is basically constant for $\Gamma \lesssim 0.1$ and starts to grow sharply beyond this threshold; for the BUMP system, $D$ is instead simply an increasing function of $\Gamma$. Also, particles in the BUMP system diffuse faster than the ones in the ABS system for large and intermediate values of $\Gamma$, but slower at the lowest accessible strain amplitudes.

The subdiffusive dynamics have been connected with the memory in particle motions for different glassy systems[31,32,47,48]. We probe the existence of such memory by considering the correlation in particle displacements at consecutive intervals of length $\Delta \gamma$, defined as

$$M(\Delta \gamma) = - \frac{\langle [x_i(2\Delta \gamma) - x_i(\Delta \gamma)] \cdot [x_i(\Delta \gamma) - x_i(0)] \rangle}{\langle [x_i(\Delta \gamma) - x_i(0)]^2 \rangle}, \qquad (1)$$

where the denominator is for normalization. This function is thus +1 if the motion is completely reversible, and zero if there is no memory. Figure 2d and e demonstrate that $M$ is significantly positive at small $\Delta \gamma$, i.e., the displacements are anti-correlated, has at small $\Delta \gamma$ a value that is smaller for the BUMP system than for the ABS system, and decays with a $\Gamma$-dependent rate. One can expect that the crossover in the MSD is related to the vanishing memory, i.e., that the dynamics becomes Markovian. To test this idea we determine both the crossover from sub-diffusive to diffusive regime (see Supplementary Fig. 5), which defines the yielding strain $\Delta \gamma_c$, and the minimal strain $\Delta \gamma_M$ at which $M(\Delta \gamma_M) = 0$. As shown in Fig. 2f, $\Delta \gamma_c$ displays a maximum at $\Gamma = 0.1$ for the ABS system, but decays monotonically with increasing $\Gamma$ for the BUMP system, compatible with the trends of $D(\Gamma)$ in Fig. 2c. Moreover, $\Delta \gamma_M(\Gamma)$ tracks $\Delta \gamma_c(\Gamma)$ very well. Hence one concludes that the creep dynamics is accompanied by significant memory and once the initial memory has been halved the system yields. This yielding occurs once the MSD has reached values $0.03 - 0.05d^2$, substantially higher than the typical cage size of hard-sphere-like systems ($0.01d^2$ for each spatial dimension[2]).

These findings allow us to make a first assessment regarding the distinction in dynamics between previous simulations and our experiment, as well as the influence of granular roughness. Previous simulations of model glasses[22] or soft-sphere jammed states[16–18,20] under cyclic shear, found a pronounced caging regime for $\Gamma \lesssim 0.1$ and no caging for $\Gamma \gtrsim 0.1$. Depending on whether the jammed system is close to the hard-particle limit[15,16,18] or highly compressed[17,18,20], caging at small $\Gamma$ can originate from either the confining effect due to the nearest neighbors, similar to the case of a dense hard-sphere packing, or the persistent contact network. The latter case, sometimes specified as reversibility[20], is however not necessary for caging. In contrast to these simulations, we find persistent sub-diffusive dynamics for all $\Gamma$ at small $\Delta \gamma$. Our results are also different from a hard-sphere marginal glass which does show a logarithmic growth in the MSD[25,49]. Thus we conclude that the presence of roughness alters radically the dynamics of granular systems in that sub-diffusion is always present. Furthermore, using two different systems, we discover that if the roughness is weak the mechanical response shows a singular point at $\Gamma \approx 0.1$ while for pronounced roughness this singularity disappears and the dynamics shows just a monotonous dependence on $\Gamma$.

Furthermore, we recall that the dynamics of granular systems depend significantly on the details of the driving protocol. For example, simulations using a fixed-volume condition have given evidence that jammed and unjammed states can coexist, resulting in a complex response as a function of both packing fraction and $\Gamma$[20,35]. In addition, it was found that the preparation history of a jammed state in an experimental soft particle system can alter the range of $\Gamma$ in which the dynamics is reversible[34]. However, since our system corresponds to the hard-particle limit at a constant pressure, we expect that these earlier findings are not relevant to the interpretation of our results.

## Dynamical heterogeneity

In order to obtain a deeper understanding of the relaxation of the system, we investigate the single particle-level dynamics. Figure 3

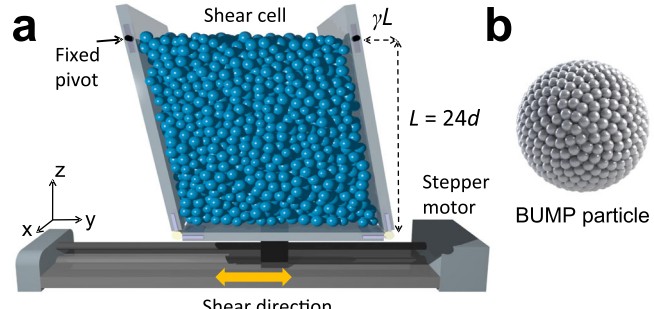

**Fig. 1 | Setup of cyclic shear experiment. a** Particles are placed into the shear cell, formed by a bottom plate attached to a stepper motor, two side plates in shear direction articulated to the bottom, and another two perpendicular side plates (not shown here). These two side plates are connected to the perpendicular plates by four pivots (two of them are shown) to maintain the shear geometry. The system is slowly sheared by driving the bottom plate by the motor. We define $L = 24d$ and the deformation $\gamma L$. X-ray tomography scans are performed at $\gamma = 0$ when the motion is stopped. **b** Visualization of a 3D-printed particle with a bumpy surface (BUMP).

shows the distributions of the particle displacements in $x$- or $y$-direction, i.e., the self-part of the Van Hove function $G_s(\delta x, \Delta\gamma)$[2], for both ABS and BUMP systems at three different $\Delta\gamma$. In each system, plotting $G_s$ for different $\Gamma$ as a function of the rescaled distance $d_x = |\delta x|/\sqrt{\langle \delta x^2 \rangle}$ roughly results in a master curve. In contrast to thermal systems[50], the shape of this curve is clearly non-Gaussian even for small $\Delta\gamma$ (panels (a) and (d); dotted lines), and there is no clear distinction between immobile and mobile particles. So the reversible particle motion is irrelevant, which is consistent with the absence of an elastic regime as mentioned above.

As found and rationalized in ref. 32, for granular systems the Van Hove function can be described well by a Gumbel law that is related to extreme value statistics,

$$G^g(d_x) = A(\lambda)\exp\left[-\frac{d_x}{\lambda} - \exp\left(-\frac{d_x}{\lambda}\right)\right], \quad (2)$$

where $\lambda$ determines the shape of the distribution and $A(\lambda)$ is a normalization factor. Figure 3 shows that this law does indeed describe well the data at small and intermediate $d_x$, i.e., $d_x \lesssim 2$ (or $d_x \lesssim 3$) for ABS (or BUMP). We find that $\lambda = 0.57$, $A(\lambda) = 2.72$ for ABS, and $\lambda = 0.59$, $A(\lambda) = 2.64$ for BUMP, i.e., the shape parameter depends only mildly on the system considered. (In ref. 32 the reported value was also close to these numbers: $\lambda = 0.605$.)

At larger $d_x$, $G_s$ displays a significant excess tail with respect to the Gumbel law, whose amplitude depends on $\Delta\gamma$ and becomes weak at $\Delta\gamma = 3\Delta\gamma_c$ for the BUMP system. The presence of such an excess shows that, in contrast to ref. 32, for the present systems the Gumbel law does not give a perfect description of the data at large $d_x$. However, this distribution can still serve as a valuable reference to discuss the shape of the real data. To quantify the strength of this excess tail over the Gumbel law, we present in Fig. 4a and b the average of the ratio $G_s(d_x)/G^g(d_x)$ in the range $d_x \in [3.5, 4.5]$ as a function of $\Delta\gamma/\Delta\gamma_c$. In general, this excess is more pronounced in the ABS system, and it decays with

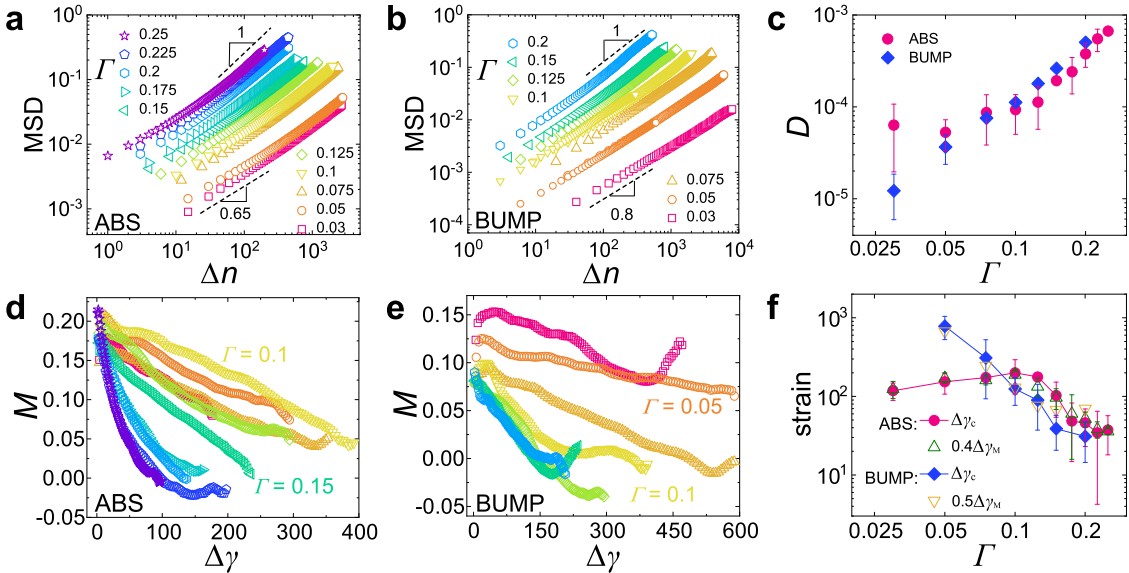

**Fig. 2 | Dynamics of cyclic shear demonstrates the absence of caging but pronounced memory effect. a**, **b** Mean squared displacement in horizontal directions vs. shear cycle number $\Delta n$ for the ABS and BUMP systems, respectively. Errors are smaller than the size of the symbols. A universal crossover from sub-diffusion (dashed lines) to normal diffusion is observed and occurs at the $\Gamma$-dependent yielding strain $\Delta\gamma_c$. Panel **b** also includes two trajectories, $\Gamma = 0.05$ and 0.1, using a higher sampling rate (factor of 5 and 10/3 respectively), to validate our measurements with a lower sampling rate. **c** $\Gamma$-dependent diffusion coefficient $D$ for the two systems. A crossover at $\Gamma \approx 0.1$ is found for the ABS system, but not for the

BUMP system. **d** and **e** Memory effect ($M$ is defined in Eq. (1)) as a function of $\Delta\gamma$ for both systems from which we define the strain $\Delta\gamma_M$ (triangles) corresponding to the vanishing of $M$. Color codes are the same as in panels **a** and **b** and errors are estimated to be 0.02. In certain cases, $\Delta\gamma_M$ is estimated from a linear extrapolation of $M(\Delta\gamma)$. **f** For both systems $\Delta\gamma_c(\Gamma)$ tracks $\Delta\gamma_M(\Gamma)$ and one observes for the ABS system again a crossover at $\Gamma \approx 0.1$. The data of $\Gamma = 0.03$ for the BUMP systems are not shown due to the poor statistics. Error bars represent the standard deviations from different realizations for ABS and fitting uncertainty for BUMP.

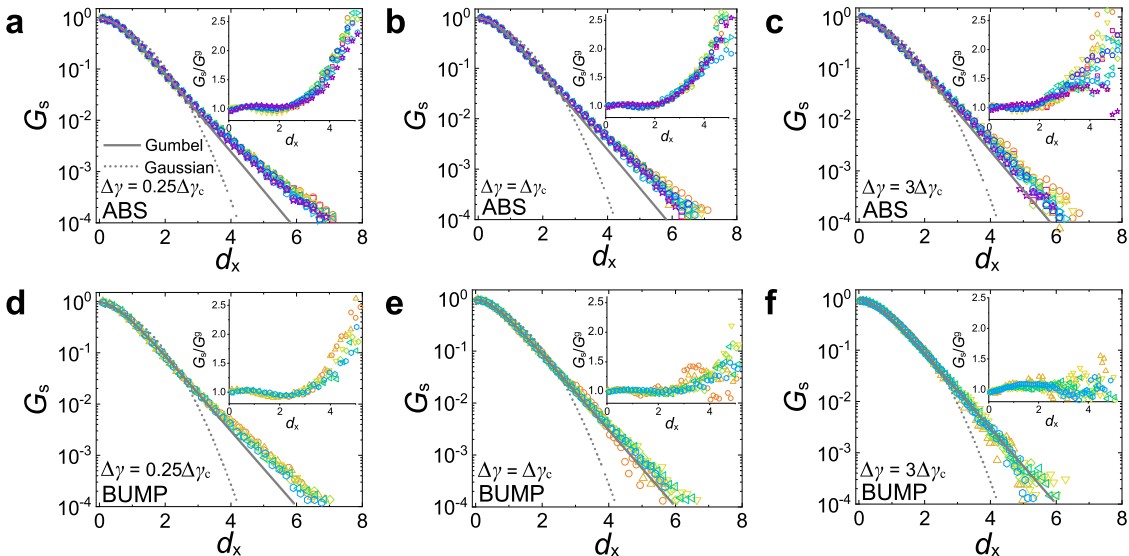

**Fig. 3 | Self-part of Van Hove function is a superposition of Gumbel law and excess exponential tail.** Panels **a**–**c** and **d**–**f** refer to ABS and BUMP systems, respectively, with the same color codes as in Fig. 2a and b, respectively. The particle displacement is expressed in terms of the normalized distance $d_x = |\delta x| / \sqrt{\langle \delta x^2 \rangle}$. For each system the data is presented for $\Delta\gamma = 0.25\Delta\gamma_c(\Gamma)$, $\Delta\gamma_c(\Gamma)$, and $3\Delta\gamma_c(\Gamma)$. Solid and dotted curves are, respectively, the Gumbel

[$G^g$, Eq. (2)] and Gaussian distributions, showing that $G_s$ is non-Gaussian even at small $\Delta\gamma$. For both systems the shape parameter $\lambda$ of the Gumbel distribution is independent of $\Delta\gamma$ and $\Gamma$, and it depends only weakly on the system (see main text). Insets show the ratio $G_s/G^g$ and one concludes that the excess exponential tail with respect to the Gumbel law becomes notable if $d_x \gtrsim 2$ for ABS and $d_x \gtrsim 3$ for BUMP.

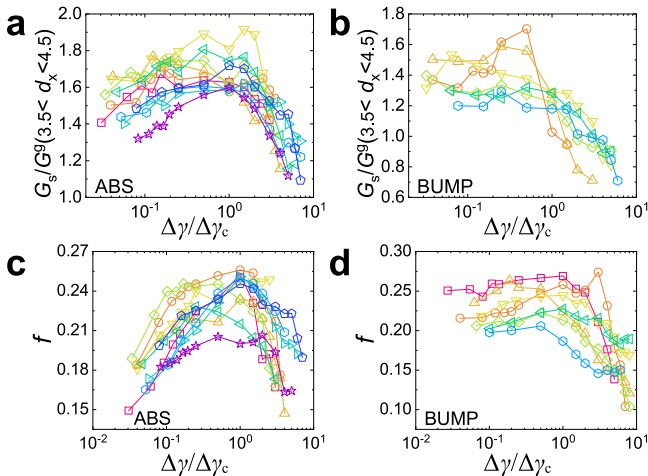

**Fig. 4 | Dynamical heterogeneity shows a peak/crossover with increasing strain for the ABS/BUMP system. a**, **b** Magnitude of the excess tail as a function of $\Delta\gamma/\Delta\gamma_c$ obtained by averaging $G_s/G^g$ in the interval $d_x \in [3.5, 4.5]$ for the ABS and BUMP systems, respectively. For the ABS system the average is larger than for the BUMP system at $\Delta\gamma \approx \Delta\gamma_c$, showing that $G_s$ has a more pronounced tail at $\Delta\gamma \approx \Delta\gamma_c$. **c**, **d** Fraction of particles involved in the largest connected cluster (two particles are defined as connected if the center distance is smaller than 1.2 times their average diameter), among the top 10% mobile particles as a function of $\Delta\gamma/\Delta\gamma_c$. Note that for the BUMP system we have set $\Delta\gamma_c(\Gamma = 0.03) = 1500$. The color codes in **a/c** and **b/d** are the same as in Fig. 2a and b, respectively.

increasing strain for both systems. We recall that a similar trend is observed for the memory, Fig. 2d, e, which is in line with the theoretical arguments that non-Markovian processes with a significant memory give rise to a pronounced tail in the Van Hove function[51]. $G_s(d_x)/G^g(d_x)$ shows a clear maximum at $\Delta\gamma \approx \Delta\gamma_c$ for the ABS system, while it decays basically in a continuous manner with increasing strain for the BUMP system. Also, $G_s(d_x)/G^g(d_x)$ decays much faster for the BUMP system and even becomes smaller than 1, i.e., the Gaussian limit is approached faster than for the ABS system. Thus these results indicate that the ABS

system with the weak surface roughness has more pronounced dynamical heterogeneities than the BUMP system.

As a direct probe of dynamical heterogeneity, we determine the spatial arrangement of the fastest particles (top 10%) and calculate the number of particles belonging to the largest connected cluster (defined via a nearest neighbor criterion). Figure 4c and d present this number $f$, after normalization by the total number of fast particles. One notices immediately that for both systems the $\Delta\gamma$-dependence of $f$ resembles that of $G_s/G^g$. This observation indicates that yielding is accompanied by maximal cooperativity for the ABS system, while such an enhanced collective phenomenon is absent for the BUMP system. For the latter, the value of $f$ is relatively high for $\Delta\gamma \lesssim \Delta\gamma_c$, i.e., during the sub-diffusive regime the dynamics are strongly cooperative. We note that a random choice of 10% of the particles in the sample gives a $f \approx 0.04$, well below the values we find here, demonstrating that the observed clustering is indeed significant. (See Supplementary Fig. 6 for the cluster size distribution.)

We quantify the evolution of the shape of $G_s$ via the non-Gaussian parameter $\alpha_2(\Delta\gamma) = \langle d_x^4 \rangle / (3\langle d_x^2 \rangle^2) - 1$, shown in Fig. 5a and b[47]. Again, we find that for both systems $\alpha_2$ shows very similar trends as the ones presented in Fig. 4. From the Gumbel law one obtains $\alpha_2 \approx 0.5$, which is substantially smaller than the observed $\alpha_2$ at small $\Delta\gamma$, which indicates that the exponential tail found in $G_s$ (Fig. 3) contributes considerably to $\alpha_2$. The $\Delta\gamma$-dependence of $\alpha_2$ can be described well by the functional form $\alpha_2(\Delta\gamma) = B(\Gamma) \exp[-(\Delta\gamma/\Delta\gamma_g(\Gamma))^\theta]$ (solid curves in Fig. 5a and b), where $\Delta\gamma_g$ is the strain scale for the recovery of Gaussian dynamics and $B$ is an amplitude, which is displayed in Fig. 5c and d. Specifically, $\alpha_2$ decays exponentially ($\theta = 1$) at $\Delta\gamma \gtrsim \Delta\gamma_c$ (i.e., after yielding) for the ABS system, while for the BUMP system, it shows a stretched exponential ($\theta = 0.5$) behavior almost from the very beginning. For both systems $\Delta\gamma_g$ closely tracks the $\Gamma$-dependence of $\Delta\gamma_c$, similar to $\Delta\gamma_M$ in Fig. 2f, i.e., a clear peak at $\Gamma = 0.1$ is observed for the ABS system. The insets present $B(\Gamma)$ and further support that dynamical heterogeneity is maximal at $\Gamma = 0.1$ for ABS, while this non-monotonicity is absent for BUMP.

Our analyses of the particle-level dynamics demonstrate that particle roughness renders the granular systems significantly different from simple glass-formers in that the dynamical heterogeneities (DH) are

already considerable even if the strain is small. Also, the Γ-dependence of the DH differs for the two systems: For weak roughness the strength of the DH peaks at $\Delta\gamma_c$, and this effect becomes most pronounced at $\Gamma \approx 0.1$, while for strong roughness no such extrema is observed.

## Yielding as a phase transition

The non-monotonic Γ-dependence of the yielding dynamics for the ABS system invites us to probe the details of these dynamics. Previous studies of simple sheared systems have given evidence that yielding can be interpreted as a point at which the system undergoes a dynamic phase transition[10,12]. While in a simple shear experiment strain completely governs the yielding dynamics, our cyclic shear setup allows to probe this phenomenon as a function of two parameters, Γ and Δγ, thus

allowing us to reach a better understanding of this feature. For this, we consider as an order parameter the so-called overlap $Q(\Delta\gamma) \in [0, 1]$, which measures the similarity of two configurations separated by Δγ (see Methods)[10]. One expects that $\langle Q \rangle$ decreases with increasing Δγ and that the shape of the distribution $P(Q)$ allows to identify the nature of the phase transition. (Note that $Q$ is a system averaged quantity, thus obtaining $P(Q)$ with good precision is a substantial experimental effort and hence we did not produce this data for the BUMP system.) Figure 6a–c shows that for the ABS system $P(Q)$ peaks close to $Q = 1$ if $\Delta\gamma/\Delta\gamma_c$ is small, i.e., most particles have not yet moved significantly. With increasing $\Delta\gamma/\Delta\gamma_c$, the peak in $P(Q)$ shifts to smaller $Q$'s before converging to the random distribution. For all Δγ considered, the width of $P(Q)$ is significantly larger for $\Gamma \leq 0.1$ than for $\Gamma = 0.2$. This is quantified in Fig. 6d by plotting $P(Q)$ at $\Delta\gamma = \Delta\gamma_c$ for different Γ, resulting in two master curves: A wide one for $\Gamma \lesssim 0.1$ and a narrow one for $\Gamma \gtrsim 0.1$. The inset of Fig. 6d confirms that the standard deviation of $P(Q)$ strongly drops at $\Gamma \approx 0.1$. This result supplements thus the information on the Γ-dependence in Figs. 4 and 5, and demonstrates that the dynamics at yielding is more cooperative if $\Gamma \lesssim 0.1$. In Fig. 6e, the Δγ-dependence of the order parameter, $\langle Q \rangle(\Delta\gamma)$, shows a non-monotonic dependence on Γ, in agreement with the trend in $\Delta\gamma_c(\Gamma)$. This non-monotonic behavior can be appreciated in the dynamic phase diagram Fig. 7a where we present $\Delta\gamma_c$ as a function of Γ, and also compare it with the case for the BUMP system (panel (b)). The reasons for these dependencies on Γ and the roughness will be discussed in the following.

The simulation study of ref. 10 showed that for a simple glass-former $P(Q)$ displays at yielding a pronounced double peak structure. This feature can be rationalized by the elasto-plastic behavior of such systems, in agreement with the caging-dynamics found if such a glass is cyclically sheared[22]. Since in our systems elasticity/caging is absent, the distribution $P(Q)$ displays a single peak, whose width however strongly depends on Γ. Whether or not this behavior can be related to an underlying dynamic phase transition cannot be decided with the present quality of the data and hence this question should be addressed in future studies.

## The potential energy landscape of granular systems

The above-presented observations on the dynamics can be rationalized qualitatively within the framework of the potential energy landscape (PEL) of the system, a view that has been very fruitful in the

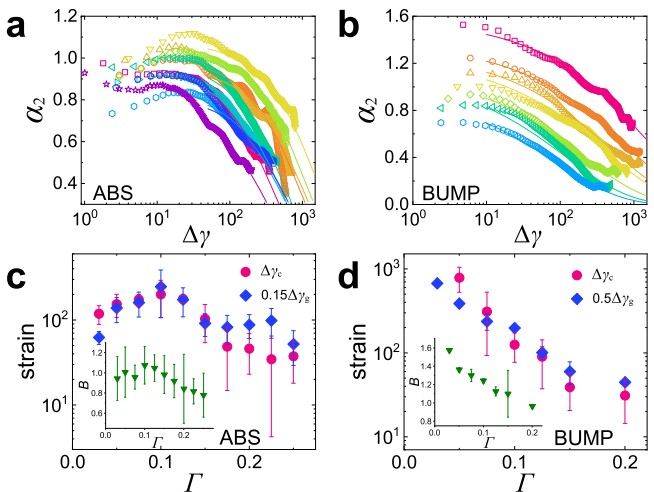

**Fig. 5 | Non-Gaussian parameter displays qualitatively different Γ − dependence for the two systems. a, b** Non-Gaussian parameter $\alpha_2$ as a function of Δγ for the ABS and BUMP systems, respectively (with the same color codes as in Fig. 2a and b). Solid curves show a fit $\alpha_2(\Delta\gamma) = B \cdot \exp[-(\Delta\gamma/\Delta\gamma_g)^\theta]$, with $\theta = 1$ (exponential) and 0.5 (stretched exponential) for ABS and BUMP, respectively. **c, d** Associated fit parameters as a function of Γ, demonstrating that $\Delta\gamma_c(\Gamma)$ and $\Delta\gamma_g(\Gamma)$ are proportional to each other. Insets: Γ-dependence of $B(\Gamma)$. Error bars represent the standard deviations from different realizations for ABS, and fitting uncertainty for BUMP.

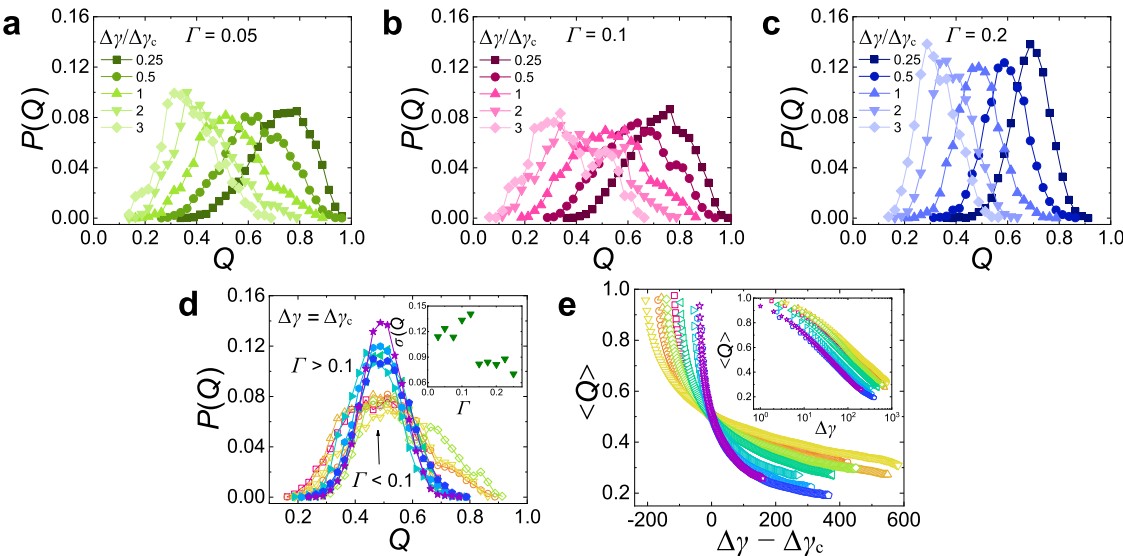

**Fig. 6 | Overlap function for the ABS system reveals the change in microscopic dynamics at yielding. a–c** Overlap function distribution $P(Q)$ as a function of $\Delta\gamma/\Delta\gamma_c$ for Γ = 0.05, 0.1, and 0.2. $P(Q)$ shifts to smaller values as $\Delta\gamma/\Delta\gamma_c$ grows (see legends). **d** $P(Q)$ at the yielding point $\Delta\gamma = \Delta\gamma_c$ shows the presence of two master

curves. Inset: The standard deviation of $P(Q)$ versus Γ has a sharp transition at $\Gamma \approx 0.1$. **e** $\langle Q \rangle$ as a function of $\Delta\gamma - \Delta\gamma_c$. The decay is slowest for $\Gamma \approx 0.1$ indicating the presence of a critical slowing down close to a critical point. Inset: Also $\langle Q \rangle$ versus Δγ shows a slowing down at $\Gamma \approx 0.1$. In **d** and **e** the color codes are the same as in Fig. 2a.

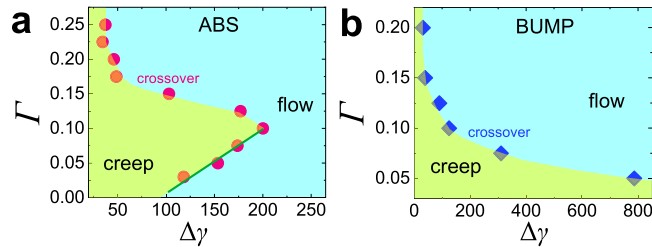

**Fig. 7 | Dynamic phase diagram of granular systems with roughness. a**, **b** Correspond to the ABS and BUMP systems, respectively. For all $\Gamma$, the system evolves with increasing $\Delta\gamma$ from sub-diffusive dynamics (yellow area) to diffusive dynamics (blue area), with the yielding point at $\Delta\gamma = \Delta\gamma_c$ (symbols). For small particle roughness (ABS) (**a**) this yielding resembles a dynamic phase transition for $\Gamma \lesssim 0.1$ (green solid line), while for larger $\Gamma$ as well as large particle roughness (BUMP) (**b**) one has only a continuous crossover.

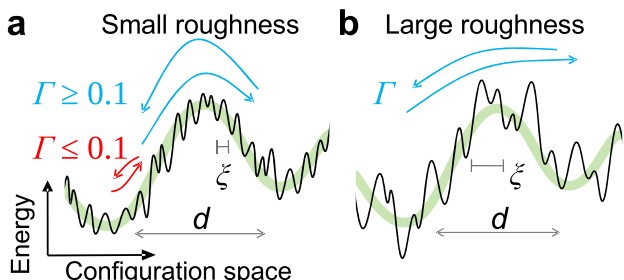

**Fig. 8 | Schematics of how a granular system explores its potential energy landscape.** Due to the particle surface roughness, the PEL has not only a structure on the particle scale $d$ (green line), but also a micro-corrugation with a characteristic size $\xi$. Starting in the left metabasin (MB) of the PEL, the double arrows indicate the back and forth motion of the system during a shear cycle. **a** The dynamics for small roughness ($\xi \ll d$) shows two regimes depending on whether the system leaves the MB during a cycle (blue arrow) or not (red arrow). **b** The dynamics for large roughness evolve smoothly with $\Gamma$ since $\xi$ becomes comparable to $d$.

context of standard glass-forming systems[52]. For such systems the PEL has an exponential number of local minima separated by a distance of the order of the particle diameter $d$ and by barriers with a height comparable with the activation energy for flow. Particle motion corresponds to the exploration of this complex landscape that can be decomposed via the barriers into basins. As shown in Fig. 8, the roughness of the particles modifies this PEL by decorating it with a multitude of additional minima and peaks, thus creating micro-basins, since roughness allows to stabilize a larger number of packings[44]. The distance between these micro-basins is not fixed but distributed with a characteristic size $\xi$ which increases with roughness.

The presence of the micro-corrugation in the PEL allows the system to move even if $\Gamma$ is small since $\Gamma \cdot d$ will always exceed the distance between some of the micro-basins, and hence caging is avoided. The system yields once the particle-scale PEL barriers (green curves in Fig. 8) have been crossed. Before this yielding, the system mainly explores the random micro-corrugation within one (particle-size) basin, thus giving rise to the subdiffusive dynamics, clearly visible in all cases when the MSD is small, Fig. 2a and b. That the dynamics of the two systems shows a very different $\Gamma$-dependence, can be rationalized by the reasonable assumption that for the ABS system the corrugation scale $\xi$ is much smaller than $d$, Fig. 8a, while this is not the case for the BUMP system, Fig. 8b. Hence, for the former system, we observe two different regimes of dynamics which are separated at $\Gamma \approx 0.1$, i.e., the typical basin size[52], while for the BUMP system the dynamics simply evolves smoothly as a function of $\Gamma$ since the difference between $\xi$ and $d$ is much less pronounced. This also explains why the subdiffusive

exponent for the BUMP system, 0.8, is much closer to 1.0 than the 0.65 found for the ABS system.

The existence of the micro-corrugation is also the reason why we observe for both systems heterogeneous dynamics at small $\Delta\gamma$, Fig. 5a and b: Before yielding the system is not rattling inside a basin like a simple glass-former but instead explores the disordered micro-corrugation. For the ABS system one can distinguish two cases before yielding: For $\Gamma \lesssim 0.1$, the system has to overcome a multitude of barriers that separate the micro-basins before it yields (after many cycles), resulting in a wide distribution $P(Q)$, Fig. 6a and b. In contrast to this, the dynamics for $\Gamma \gtrsim 0.1$ are much less affected by the presence of these micro-basins since at each cycle the system will cross many of them, leading to a more narrow distribution $P(Q)$, Fig. 6c. Such a $\Gamma$-dependence can be expected to be absent for the BUMP system because the condition $\xi \ll d$ no longer holds. For the same reason, we observe peak heterogeneity at yielding for the ABS system (Figs. 4a/c and 5), but not for the BUMP system.

## Discussion

Under simple shear conditions, yielding is associated with a drop or crossover in the stress-strain curve, signaling the transition from elastic to plastic behavior. For this type of driving, the mechanical response of granular materials can be expected to be very similar to one of the thermal glass-formers since the PEL's micro-corrugation is basically irrelevant[1]. In contrast to this, the cyclic shear considered here permits to investigate the highly non-trivial dynamic consequences of this micro-corrugation, such as the identified creep behavior (i.e., a pronounced sub-diffusion for all $\Gamma$) in a steady state. This creep motion can hence be considered to be a hallmark of granular materials. However, earlier simulations of smooth particles, e.g.[16], did not observe the sub-diffusive dynamics. Therefore we argue that this type of dynamics is due to the PEL's micro-corrugation which originates from the surface roughness of particles, intrinsic for real granular materials. A comparison between the ABS and BUMP systems demonstrates the significant influence of roughness on the dynamics as well as the underlying PEL, i.e., the surface roughness serves as a control parameter of the topography of the PEL and thus the nature of yielding. Further studies, in which roughness is systematically varied, will be helpful to elucidate the details of the observed creep dynamics.

In line with the creep motion, the evolution of the packing fraction during a single shear cycle displays persistent hysteresis for different $\Gamma$, see Supplementary Fig. 7, similar to the shear stress[45]. This is in contrast to the energy and stress of glasses which show clear reversibility for small $\Gamma$[24], thus supporting the view that granular systems and glasses are different.

Our results are relevant for a multitude of situations in geoscience[7], such as landslide processes which are preceded by complex external disturbances, or small tremors that will affect the long-time stability of civil engineering structures such as dams. Interestingly, a recent experiment reported the persistent creep of a sandpile without an apparent external disturbance[53], which indicates that the particle-level relaxation can be triggered by multiple microscopic processes. This finding is thus in line with our results, in that we reveal the absence of caging even if $\Gamma$ is very small. Hence granular materials have no well-defined stability threshold, in stark contrast to other amorphous solids.

We also mention that for simple shear it is custom to classify the yielding as either ductile or brittle. Our results show that for granular materials the nature of yielding depends strongly on the driving protocol, i.e., simple shear vs. cyclic shear, and also on the shear amplitude $\Gamma$. This dependence, which is absent in more standard disordered materials, suggests that for granular materials such a classification might not be possible. Moreover, the impact of other parameters, e.g., particle shape, on the granular dynamics is at present not known. Advancing on these points will lead to a fundamental understanding of

granular rheology and thus permit to develop a new holistic view of the failure of complex materials.

## Methods
### Experiment
We consider two types of spherical granular beads: (1) Acrylonitrile butadiene styrene plastic (ABS) and (2) 3D-printed (ProJet MJP 2500 Plus, 0.032 mm resolution) with a bumpy surface (BUMP). Both systems are bidisperse (50:50 composition) with diameters 5 mm and 6 mm. The unit length is $d = 5$ mm. A BUMP particle is created by uniformly decorating the surface of a central sphere with 400 half-spheres of diameter $\approx 5\%d$[44], shown in Fig. 1b. The corresponding random loose packing fractions are $\phi_{\mathrm{RLP}} \approx 0.61$ for ABS and $\phi_{\mathrm{RLP}} \approx 0.58$ for BUMP, which confirms that the BUMP particles are indeed rougher than the ABS beads.

The set-up of the cyclic shear cell is presented in Fig. 1 and described in the caption. Shear is induced by a stepper motor and the strain rate is $\dot{\gamma} \approx 0.13/s$, giving a dimensionless inertia number $I = \dot{\gamma} d \sqrt{\rho/P} \approx 6 \times 10^{-4}$, where we estimate the pressure to be given by $P \approx \rho g L$ and $L \approx 24d$. (Here $\rho$ is the mass density of the particles.) This value corresponds to a quasistatic shear condition[46]. The considered range of cyclic shear amplitudes is $\Gamma = 0.03 - 0.25$.

After the beads have been deposited in the cell (i.e., having reproducible $\phi_{\mathrm{RLP}}$), we start to shear the system with a given $\Gamma$, and the packing displays a transient compaction process, see Extended Data Fig. 1. Despite the different roughness and $\phi_{\mathrm{RLP}}$ for the ABS and BUMP systems, their steady-state packing fractions are approximately the same. After a sufficient number of shear cycles, depending on $\Gamma$ typically 1000−5000, we start to collect data for the steady state dynamics. These are the results presented in the main text.

X-ray tomography scans (UEG Medical Group Ltd., 0.2 mm spatial resolution) are performed periodically when the cyclic shear is completed (shearing motion is stopped). For each scan, the acquired data are analyzed using custom-made image processing codes that allow locating the center of a particle center to within $0.001d$, thus constructing the 3D packing. Depending on $\Gamma$, the sampling period is between 1 and 40 cycles, ensuring unambiguous particle tracking. For a given $\Gamma$, we perform a sequence of 150−200 scans, forming a single trajectory. Each individual particle is tracked by requiring that its displacement between two consecutive scans is smaller than $0.5d$. (We find that nearly all the particles can be tracked unambiguously, except for the very few near the boundary which had a slightly faster motion and errors in image processing.) We analyze the dynamics after excluding the particles within $4d$ from the boundary, which leaves us with 3500−4000 particles.

### Overlap function
For a system with $N$ particles the overlap function, which quantifies the similarity of two configurations (here separated by a strain $\Delta\gamma$), is defined as

$$Q(\Delta\gamma) = \frac{1}{N}\sum_{i=1}^{N}\Theta(c - |\delta x_i(\Delta\gamma)|), \qquad (3)$$

where $\delta x_i$ is the particle displacement, $\Theta(\cdot)$ is the Heaviside step function, and $c$ is a preset threshold[10]. By definition, $0 \leq Q \leq 1$, and $Q$ decreases as the system moves away from its initial configuration. In practice, we divide the cubic probe space into $2 \times 2 \times 2$ non-overlapping subsystems, each having $N \approx 500$, to increase the number of measurements of $Q$. Then, for given $\Gamma$ and $\Delta\gamma$, $Q$ is sampled from different subsystems, starting configurations, $x$-and $y$-directions, as well as 3−5 independent realizations. To calculate $Q$ we choose $c = 1.15\sqrt{\langle \delta x^2(\Delta\gamma_c)\rangle}$, which makes that $\langle Q(\Delta\gamma_c)\rangle = 0.5$. This threshold must be chosen to depend on $\Gamma$, since the MSD in the subdiffusive regime changes strongly with $\Gamma$, see Fig. 2a.

## Data availability
The data that support the findings of this study are available from https://zenodo.org/records/10963940 or from the corresponding authors.

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

## Acknowledgements

We are grateful to S. Burov for the discussions. This work is supported by the National Natural Science Foundation of China (No. 11974240), and the Science and Technology Commission of Shanghai Municipality (No. 22YF1419900). Y. Y. acknowledges support from the fellowship of the China Postdoctoral Science Foundation (No. 2021M702151). W. K. is a senior member of the Institut Universitaire de France.

## Author contributions

Y.Y., Y.W., and W.K. designed the research. Y.Y., Z.Z., Y.X., H.Y., and S.Z. performed the experiment. Y.Y., W.K., and Y.W. analyzed the data and wrote the paper.

## Competing interests

The authors declare no competing interests.
