## [Peer Review File · Nature Communications]

From creep to flow: Granular materials under cyclic shearEditorial Note: Parts of this Peer Review File have been redacted as indicated to remove third-party material where no permission to publish could be obtained.

REVIEWER COMMENTS

Reviewer #1 (Remarks to the Author):

The paper by Ye Yuan et al. investigates the cyclic shear of a granular suspension by x-ray tomography, providing insight into the internal dynamics, memory, overlap function and dynamic heterogeneity. From the observed microscopic behavior, the authors conclude that (i) the observed creep at small strain amplitude is very different from the close-to-elastic response of other glassy materials, (ii) the dynamics are most cooperative at the yielding transition and (iii) the nature of the transition changes from small to large strain amplitude. While the results are interesting, and the measurements likely performed with care and accuracy (although this is not clear from the text as the measurements and errors are not discussed at all), the text is quite unclear: this appears to be due to lack of proper description of the results, as well as a serious mixing of the author's interpretation (not sufficiently supported by the data) and their results. Nevertheless, due to the importance of the results and the potentially significant insight that can be gained from the system, I think it is worth considering a major revision that takes into account my comments below.

1. The lack of information starts immediately at the beginning of the experimental description, paragraph "Cyclic shear experiment". No sketch of the system is provided (which is important to understand the "horizontal" "x-y" displacements later, are those in the direction of the applied shear or perpendicular?), and no time scales (what is the time scale of a shear cycle, and how does it compare to the acquisition time; is the system stopped during acquisition?) and no error bars (what's the error bar of the particle location? Can all particles always be tracked over shear though pictures are only taken after each cycle, and the MSD in Fig. 1a goes up to almost 1, meaning 1 particle diameter, at which it would be impossible to track the particles). Better description is needed to understand and judge the results.

The unclear description continues in the results section, where also a mixing of results and interpretation adds to the confusion:

2. Page 4: While I agree that the remaining MSD slope (and absence of a plateau) at small Δn indicates the absence of an elastic regime (caging), the conclusion that this is due to micro-corrugation of the granular particles is unclear to me and not supported by the data. Have the authors tried different surface roughness and observed a systematic change of the creep behaviour? I'm doubtful because also (real) colloidal particles have rough surface; either due to porosity of the particles or due to the sterically stabilizing, grafted ligands on the surface, and this surface roughness is known to give rise to bending rigidity in colloidal gels. An alternative explanation of the different behavior (creep) at small strain could also be the absence of thermal motion, preventing local equilibration of forces by rattling. Since the authors present this corrugation (and the corresponding energy landscape) as underlying reason for some of the granular behavior, they need to provide more evidence for this, e.g. in terms of a systematic behavior change when changing the roughness.

3. Paragraph from line 115 to 129 is difficult to understand, among others because interpretation and results are mixed. Fig. 1a and c does NOT show how the metabasin is

explored, and it does NOT show that the yielding is highly collective. All I'm seeing is a bunch of MSD curves that have a crossover, and a diffusion coefficient that increases slightly for $\Gamma < 0.1$, and increases more strongly thereafter. The authors have to describe much more accurately what they observe and what they hypothesize to be happening. Why can a particle leave the metabasin for $\Gamma > 1$ for each cycle? For a particle to move across another particle into the next minimum, it needs a strain of order ~ 1 . The whole paragraph needs major rewrite.

4. Paragraph from line 145 to 157 difficult to read as the wording is inaccurate. For example the authors say that "Figures 2a-c show that $P(Q)$ are close to one". But I believe what they mean is not that P is close to 1, but rather that Q (i.e. the peak of the $P(Q)$ distribution) is close to 1. Also thereafter: "With increasing ... $P(Q)$ shifts to the left". I believe what they mean is that the peak in the $P(Q)$ distribution shifts to the left etc. These inaccuracies in writing make the text difficult to follow.

5. The paragraph thereafter, line 158 to 176, I find misleading and overinterpreted. The authors conclude from the lack of a double peak as observed in ref. 8 that their system exhibits a weak first order transition. This is completely unsupported. The system in ref. 8 is very different, and from the presence of just one peak in the present granular system, the authors cannot conclude anything, neither a first nor a second order transition of any kind, it simply doesn't allow to reach any conclusion. Then, speaking of a surface tension in this context goes way to far and is not understandable. Connecting that to the micro-corrugation of the particles is neither supported. The authors should strictly stick to their data, and delineate clearly what they can deduce from it.

6. The paragraph about dynamic heterogeneity to me is the most credible, but I'm not sure why the distributions are fitted by a Gumbel law. Is it supposed to be fitted by this law and why? What's the physical motivation for it? I understand (non-)Gaussianity, but I cannot place the Gumbel law. Physical reasoning should be provided.

7. Conclusions: Following my previous doubts, I think some of the drawn conclusions are simply not supported by the data and would need more evidence. For example the relevance of the micro-corrugation of the granular particles. The authors say: "The details of the yielding dynamics, i.e., the nature of the phase transition, will depend on the micro-roughness, shape, as well as the friction coefficient of the particles, since all these parameters influence the micro-corrugation of the PEL and hence the system dynamics." Could be, but they haven't shown this. If this is in the conclusions, supporting evidence (i.e. varying particle roughness) is needed in the manuscript.

Reviewer #2 (Remarks to the Author):

The manuscript describes results of cyclic shear experiments of granular matter, which seems to suggest that the reorganization of the system involves a crossover from creep to diffusive motion. This is in contradiction of a fairly large body of literature, including granular matter, which shows that there is a transition between absorbing (or cyclic) states and diffusive states. This would warrant the paper being taken seriously as presenting a new result, but unfortunately, the manuscript does a very poor job of making its case. Specifically, the results contradict the work of Royer and Chaikin (with the paper

title 'Precisely cyclic sand' making it abundantly clear the disagreement) and the authors do not make any effort to explain why their results should be accepted as an improvement of that of Royer and Chaikin. I note some points below where I found the manuscript inadequate. My overall assessment is that the manuscript really needs to present a much stronger and tighter case than what is currently done in order for it to be considered further.

Experimental set up needs a better description. Since the authors report on compaction, it appears that the cell volume is not held fixed but the details of how it is allowed to vary is not described.

The authors invoke a potential energy surface while at the same time discussing the rough surfaces of the grains. Does the latter imply a role of friction? If so, speaking of a PEL is unclear. The authors should clarify.

In cyclically sheared systems, there are prolonged transients and typically the behaviour is reported in the steady state. Since the authors do not discuss this in any detail at all, it is not clear what part of the results in Fig. 1 are related to transient behaviour, and what is steady state behaviour. Fig. 1 a shows an MSD that always has finite slope dependence on the number of cycles, and in this case, there is no precisely defined yield point. The authors appear to be stating that at long times there is always a diffusive regime. Then this is not creep (it is not creep even if the dependence is a fractional power law). The observed maximum in what the authors confusingly call the yielding strain is consistent with earlier observations, which report on the duration of the transient to reach the steady state, but the effect here is much weaker.

Reviewer 1 (Remarks to the Author):

The paper by Ye Yuan et al. investigates the cyclic shear of a granular suspension by x-ray tomography, providing insight into the internal dynamics, memory, overlap function and dynamic heterogeneity. From the observed microscopic behavior, the authors conclude that (i) the observed creep at small strain amplitude is very different from the close-to-elastic response of other glassy materials, (ii) the dynamics are most cooperative at the yielding transition and (iii) the nature of the transition changes from small to large strain amplitude. While the results are interesting, and the measurements likely performed with care and accuracy (although this is not clear from the text as the measurements and errors are not discussed at all), the text is quite unclear: this appears to be due to lack of proper description of the results, as well as a serious mixing of the author’s interpretation (not sufficiently supported by the data) and their results. Nevertheless, due to the importance of the results and the potentially significant insight that can be gained from the system, I think it is worth considering a major revision that takes into account my comments below.

We thank the referee for judging our results to be interesting. We acknowledge that part of the text might not have been optimal and therefore we have reorganized and rewritten a very large part of the manuscript. More details are given below.

1. The lack of information starts immediately at the beginning of the experimental description, paragraph “Cyclic shear experiment”. No sketch of the system is provided (which is important to understand the “horizontal” “x-y” displacements later, are those in the direction of the applied shear or perpendicular?), and no time scales (what is the time scale of a shear cycle, and how does it compare to the acquisition time; is the system stopped during acquisition?) and no error bars (what’s the error bar of the particle location? Can all particles always be tracked over shear though pictures are only taken after each cycle, and the MSD in Fig. 1a goes up to almost 1, meaning 1 particle diameter, at which it would be impossible to track the particles). Better description is needed to understand and judge the results.

We thank the referee for raising these points. We have now added Fig. 1 which shows the experimental setup. The strain rates are now given in the Methods, 0.13/s, from which one can obtain the time scales needed to make one cycle. (Note that this time depends on the strain amplitude Γ .) As we show in the Methods section, this corresponds to a quasi-static

deformation. We now also state in the Methods that we stopped the shearing periodically (at strain zero) to make the CT scan and also give the error bars of the particle positions ($10^{-3}d$).

The motion of the particles is indeed sufficiently slow that we can track all of them. Note that the fact that the MSD becomes $O(d)$ does not mean that a particle has made a jump of size d since this value of the MSD is attained only after many cycles. So there is no problem that particles are “lost”, as we now state in the Methods.

Modification to the manuscript:

-Added Fig. 1 and Methods section

-Details on strain rate (line 424)

-Information on halting cyclic motion (caption of Fig. 1 and line 435)

-Information on positional resolution (lines 84 and 437)

The unclear description continues in the results section, where also a mixing of results and interpretation adds to the confusion:

2. Page 4: While I agree that the remaining MSD slope (and absence of a plateau) at small Δn indicates the absence of an elastic regime (caging), the conclusion that this is due to micro-corrugation of the granular particles is unclear to me and not supported by the data. Have the authors tried different surface roughness and observed a systematic change of the creep behaviour? I’m doubtful because also (real) colloidal particles have rough surface; either due to porosity of the particles or due to the sterically stabilizing, grafted ligands on the surface, and this surface roughness is known to give rise to bending rigidity in colloidal gels. An alternative explanation of the different behavior (creep) at small strain could also be the absence of thermal motion, preventing local equilibration of forces by rattling. Since the authors present this corrugation (and the corresponding energy landscape) as underlying reason for some of the granular behavior, they need to provide more evidence for this, e.g. in terms of a systematic behavior change when changing the roughness.

We thank for the referee for bringing up this important point. In order to probe the influence of the surface roughness we have now determined the dynamics of a second system: Particles with a very pronounced roughness, i.e., 5% of their diameter. (Carrying out these new experiments and analyzing the data took quite a bit of time and this is the reason why we submit the revised version of the manuscript only now.) We find that also this second

system shows a sub-diffusive dynamics at small and intermediate (accumulated) strain, which demonstrates that this is a robust feature for the dynamics of granular systems. Interestingly, however, the strain amplitude dependence of many other properties differ from the ones found for the previously studied system. The combination of our results with simulation data from the literature allows us now to obtain a comprehensive view of the relaxation and yielding dynamics of granular systems. This insight can be expressed in a simple and natural manner using the geometrical properties of the potential energy landscape (PEL) of the system (new section “The Potential Energy Landscape of granular systems” and new Fig. 7).

Regarding the fact that also colloidal systems can have rough surfaces, caused by the internal structure of the particles or by molecules at their surfaces: This is certainly correct. However, the nature of the particle interactions in such systems is quite complex since it involves local charges, steric and bonding interactions with the grafted ligands/polymers, interaction via the solvent, etc. So at the end it is not obvious that on the mesoscopic length scale of the colloidal particles the effective interaction between the particles can indeed be considered as a rough hard sphere (i.e., only repulsion). As a consequence it must be expected that the potential energy landscape of colloidal systems is way more complex than the one we have for a granular material which, having macroscopic size particles, can be considered to be close to a system with ideal hard-core interactions. So we think that colloidal systems that have a rough surface are not a good analogue for granular materials and in fact we are not aware of experimental data for such systems that show, *in oscillatory steady state conditions*, a sub-diffusive dynamics. We have also asked our colleagues in the soft matter community about this and nobody could give us an example for such a behavior.

It is in fact quite likely that this absence of a sub-diffusive dynamics in colloidal systems is due to the second point that the referee makes: Colloidal systems do have thermal motion while the granular systems do not, i.e., under quasistatic driving granular systems always stay mechanically stable, i.e., at zero temperature. As a consequence the dynamics of colloids is not hindered by the micro-corrugation of its PEL (the system is floating above it), while the granular system is forced to explore the surface of the PEL, i.e., it will be affected by the micro-structure of the PEL. This might thus be the explanation why on small length scales the dynamics of the two types of systems are so different. Furthermore we emphasize that this sub-diffusive dynamics is not related to out-of-equilibrium effects (a possibility

evoked by the second referee) since we are at steady state conditions.

Modification to the manuscript:

-Added the new results on the second system

-Moved all the discussion regarding the connection between the dynamical features and the PEL to a new section at the end of the manuscript

3. Paragraph from line 115 to 129 is difficult to understand, among others because interpretation and results are mixed. Fig. 1a and c does NOT show how the metabasin is explored, and it does NOT show that the yielding is highly collective. All I'm seeing is a bunch of MSD curves that have a crossover, and a diffusion coefficient that increases slightly for $\Gamma < 0.1$, and increases more strongly thereafter. The authors have to describe much more accurately what they observe and what they hypothesize to be happening. Why can a particle leave the metabasin for $\Gamma > 0.1$ for each cycle? For a particle to move across another particle into the next minimum, it needs a strain of order 1. The whole paragraph needs major rewrite.

We agree with the referee that our discussion regarding the connection between the metabasins and the dynamics was not sufficiently clear. As a consequence we have completely rewritten this section. In particular we now focus on the discussion of the data (including the new system) and postpone the interpretation/implication of the results to the last part of the manuscript, where we have added a new section how the PEL can help to understand the various features of the relaxation dynamics of the two granular systems. This new section comes only after all the "raw" data has been presented and thus we avoid the mixing of real data and interpretation.

Regarding the fact that for $\Gamma \geq 0.1$ the particles can leave a metabasin: Also this part of the text was not sufficiently clear. In the new version of the text we have simplified the chain of arguments and in particular avoided to make reference to metabasins. However, already here we want to emphasize that many previous studies have shown that under simple shear the system yields once the strain has reached a value around 0.1, i.e., only a small fraction of the size of the particles. This small value is related to the fact (discussed in many earlier publications) that the distance between two neighboring local minima of the PEL is indeed

significantly smaller than the size of a particle.

Modification to the manuscript:

-Avoided to mix presentation of results with interpretation.

-Added discussion in which we compare the “raw” data regarding the dynamics of the two systems and also simulation data from the literature.

-Added a new section at the end of the manuscript where we discuss the interpretation of the results in the context of the PEL.

4. Paragraph from line 145 to 157 difficult to read as the wording is inaccurate. For example the authors say that “Figures 2a-c show that $P(Q)$ are close to one”. But I believe what they mean is not that P is close to 1, but rather that Q (i.e. the peak of the $P(Q)$ distribution) is close to 1. Also thereafter: “With increasing ... $P(Q)$ shifts to the left”. I believe what they mean is that the peak in the $P(Q)$ distribution shifts to the left etc. These inaccuracies in writing make the text difficult to follow.

The referee is right on both points. We now have completely rewritten this section and thus avoided the misleading information.

Modification to the manuscript:

-Rewrote the section in which we present our findings on the overlap.

5. The paragraph thereafter, line 158 to 176, I find misleading and overinterpreted. The authors conclude from the lack of a double peak as observed in ref. 8 that their system exhibits a weak first order transition. This is completely unsupported. The system in ref. 8 is very different, and from the presence of just one peak in the present granular system, the authors cannot conclude anything, neither a first nor a second order transition of any kind, it simply doesn't allow to reach any conclusion. Then, speaking of a surface tension in this context goes way to far and is not understandable. Connecting that to the micro-corrugation of the particles is neither supported. The authors should strictly stick to their data, and delineate clearly what they can deduct from it.

We thank the referee for bringing up this point. We agree that the system of Ref. [8], a thermal glass-former, is very different from ours, but the main point of our text was not to make a one-to-one comparison of that system with our system but instead use the method proposed in Ref. [8] to elucidate the motion of the particles. We also acknowledge that the

presented data do not provide strong evidence that for our system yielding corresponds to a phase transition and that the order of such a transition changes with Γ . A significantly better statistics is needed to advance on this question and therefore we have removed now that part of the discussion. However, our data is sufficiently good to see that the width of $P(Q)$, the distribution of the overlap, does change in an abrupt manner with Γ , see (present) Fig. 6(d). Hence we focus in the new version of the manuscript on this result, which is robust, as well as on the Γ -dependence of the accumulated strain needed to reach this yielding point, see (present) Fig. 7.

Modification to the manuscript:

- Completely rewrote the section in which we present our findings on the overlap.

6. The paragraph about dynamic heterogeneity to me is the most credible, but I'm not sure why the distributions are fitted by a Gumbel law. Is it supposed to be fitted by this law and why? What's the physical motivation for it? I understand (non-)Gaussianity, but I cannot place the Gumbel law. Physical reasoning should be provided.

We thank the referee for this question. Previous studies of the relaxation dynamics of granular systems have found that the Gumbel law gives indeed a good description of the Van Hove function, i.e., the distribution of the particle displacements. In Ref. [Kou et al., *Nature* **551**, 360 (2017)] it was found that this functional form describes the r -dependence of $G_s(r, t)$ over many decades (for a system of ellipsoids) and it was argued that this is due to the fact that the Gumbel law is one of the functional forms that describe extreme value statistics, i.e., the tail of distribution functions. (This motivation for making reference to the Gumbel law was obviously not emphasized enough in the previous version of the manuscript, an omission that we have now corrected.) However, if the dynamics has substantial memory, as it is the case in the system considered here, the applicability of the extreme value statistics is no longer guaranteed and thus one can expect deviations from the Gumbel law. However, this law is still a useful reference distribution since its functional form gives at small displacements a much better description of the Van Hove function than, e.g., a Gaussian. Therefore we have chosen to use this law to discuss the magnitude of the wing found in the Van Hove function at large displacement.

Modification to the manuscript:

- Rewrote the section in which we present our findings on the dynamical heterogeneity and

in particular we now motivate why we use the Gumbel distribution to discuss the shape of the Van Hove function.

7. Conclusions: Following my previous doubts, I think some of the drawn conclusions are simply not supported by the data and would need more evidence. For example the relevance of the micro-corrugation of the granular particles. The authors say: “The details of the yielding dynamics, i.e., the nature of the phase transition, will depend on the micro-roughness, shape, as well as the friction coefficient of the particles, since all these parameters influence the micro-corrugation of the PEL and hence the system dynamics.” Could be, but they haven’t shown this. If this is in the conclusions, supporting evidence (i.e. varying particle roughness) is needed in the manuscript.

Motivated by the remarks of the referee we have carried out experiments for an additional granular system that has a significantly larger surface roughness. By comparing the relaxation dynamics of these two systems one clearly sees that the roughness strongly affects this dynamics. Therefore the statement we made in the previous version of the manuscript is now supported by real data and, as we point out in the manuscript, is also compatible with the existent simulation data. It is clear that we have not explored all the possible connections between relaxation dynamics and particle roughness, but the goal of the present manuscript was to point out that roughness is a parameter that is very important for the relaxation dynamics of granular system. Figuring out the exact nature of these connections will be left for the future.

We thank the referee for his/her constructive comments that have allowed us to improve the quality of the manuscript by presenting a significantly more extended set of data.

Reviewer 2 (Remarks to the Author):

The manuscript describes results of cyclic shear experiments of granular matter, which seems to suggest that the reorganization of the system involves a crossover from creep to diffusive motion. This is in contradiction of a fairly large body of literature, including granular matter, which shows that there is a transition between absorbing (or cyclic) states and diffusive states. This would warrant the paper being taken seriously as presenting a

new result, but unfortunately, the manuscript does a very poor job of making its case. Specifically, the results contradict the work of Royer and Chaikin (with the paper title 'Precisely cyclic sand' making it abundantly clear the disagreement) and the authors do not make any effort to explain why their results should be accepted as an improvement of that of Royer and Chaikin. I note some points below where I found the manuscript inadequate. My overall assessment is that the manuscript really needs to present a much stronger and tighter case than what is currently done in order for it to be considered further.

We thank the referee for agreeing that the results presented in the manuscript have the potential “to be taken seriously”. Of course we are sorry that the referee thinks that the previous version of the manuscript did not make a strong case for showing the novelty of the results. We acknowledge that we might not have made a sufficient effort to put our findings in perspective to previous results. In view of this criticism we have strongly modified the contents of the manuscript, adding new experimental data and adapted the discussion.

We agree with the referee that there is indeed a substantial body of literature in which people have documented a transition between absorbing states and diffusion (and we think/hope that we have identified and cited the relevant works). However, we point out that most of these results stem from simulations of rather simple model systems. One finds soft sphere or Lennard-Jones-like systems, i.e., models for glass-forming systems that do show an elastic behavior at small deformations because the potential energy can be approximated by a harmonic potential, or hard sphere systems in which one finds the same behavior for the free energy. Although the authors of some of these papers claim that their simulations describe granular materials, the question is open whether all relevant ingredients have indeed been included in these simulations.

We note that most of these simulations do not take into account friction, a property that is most important for the static and dynamic behavior of *real* granular materials. The paper by Royer & Chaikin mentioned by the referee is a nice example which documents that friction can radically alter the relaxation dynamics. We are aware of only very few simulations in which the influence of roughness has been probed (and Royer & Chaikin do not consider roughness either), but these studies have not probed the relaxation dynamics, [e.g., Papanikolaou et al., PRL **110**, 198002 (2013); Ikeda et al., PRL **124**, 208001 (2020); Sun et al., PRL **125**, 268005 (2020)]. (We emphasize that we want to distinguish friction and roughness: The former is a material property while the latter is related to the shape of the

particle on the meso and macroscopic scale. Whether or not one can introduce an effective friction coefficient that takes into account both effects is an open problem in granular matter science that should be addressed in the future.)

We emphasize that we do not claim that the results of Royer & Chaikin are wrong or need to be “improved”, we just argue that in real systems other factors play an important role, and roughness is one of them. Also we emphasize that the sub-diffusive behavior has been found in other experiments of sheared granular materials [Marty et al., PRL **94**, 015701 (2005); Kou et al., Nature **551**, 360 (2017)] and hence can be regarded as a robust feature of such systems. Hence it is important to probe the dynamics of real granular systems, especially to see how roughness affects the dynamics.

Modification to the manuscript:

-We have now expanded significantly the discussion regarding the connection of our results with the results from computer simulations and previous experiments. In particular we point out that there are no conflicting results but that instead granular systems have dynamical features that depend on parameters which have so far not received sufficient attention. As a result we think to have reached a quite comprehensive view on regarding the importance of roughness for the dynamics of granular materials.

Experimental set up needs a better description. Since the authors report on compaction, it appears that the cell volume is not held fixed but the details of how it is allowed to vary is not described.

We thank the referee for pointing out this omission. We have now added a schematic in which the experimental setup is shown, Fig. 1. One recognizes that with this setup the cell volume is indeed not fixed, and allows the following two types of volume change: (1) During each shear cycle, the granular packing is dilated/compacted when a shear strain is applied/canceled; (2) Starting from the initial loose packing, the packing at $\gamma = 0$ will be compacted with applied cyclic shear. In the Extended Data Fig. 1 we present the second process. For the results presented in the main part of the manuscript we always focus on the steady state dynamics after the compaction is finished.

Modification of the manuscript:

-Added Fig. 1 and Methods section describing the experimental setup.

The authors invoke a potential energy surface while at the same time discussing the rough surfaces of the grains. Does the latter imply a role of friction? If so, speaking of a PEL is unclear. The authors should clarify.

We thank the referee for this interesting and important comment. In the present manuscript we probe how the roughness of the particles influences the dynamics of a granular system under quasistatic cyclic shear. We argue that this roughness induces a micro-corrugation on the PEL since roughness allows the existence of many local mechanically stable configurations that are unstable in the absence of roughness. This micro-corrugation is a real property of the PEL, since it is directly related to the interactions between the (rough) particles.

It is now of course a very interesting question of friction comes into play. We emphasize that friction and roughness are different quantities. In principle one can have frictionless rough particles but also particles that are perfectly smooth down to the nanoscale but that have friction. Although friction and roughness influence certain macroscopic properties of granular materials in a similar manner (decrease in packing fraction, slower relaxation dynamics,...), their differences in structure and dynamics on the particle level have not yet been well understood. The main focus of the current manuscript is the influence of the roughness of the particles on the yielding dynamics. If friction can be neglected, the PEL is a perfectly well defined concept since it depends only on the interactions between the particles. Royer & Chaikin have shown in their simulation study of smooth particles that friction affects the dynamics qualitatively, but that the phenomenon of intermittent caging is always present, i.e., there is no sub-diffusive regime. Instead our experiments show that rough particles do show such a regime (also found in previous studies [Marty et al., PRL **94**, 015701 (2005); Kou et al., Nature **551**, 360 (2017)]) and that the exponent of this sub-diffusion depends on the roughness. Therefore these results indicate that the persistent sub-diffusion is mainly caused by roughness and not by friction. It is therefore reasonable to discuss our findings within the framework of the PEL. We expect, however, that if friction becomes *very* large, that also friction will affect the yielding dynamics of granular systems, but details on this are beyond the scope of the present manuscript and thus should be addressed in the future.

Modification of the manuscript:

-Added new text in which we argue that based on our results and previous simulation data the influence of small and intermediate friction on the PEL can be neglected with respect of the one of roughness. Lines 132-143, 296-301.

In cyclically sheared systems, there are prolonged transients and typically the behaviour is reported in the steady state. Since the authors do not discuss this in any detail at all, it is not clear what part of the results in Fig. 1 are related to transient behaviour, and what is steady state behaviour. Fig. 1 a shows an MSD that always has finite slope dependence on the number of cycles, and in this case, there is no precisely defined yield point. The authors appear to be stating that at long times there is always a diffusive regime. Then this is not creep (it is not creep even if the dependence is a fractional power law). The observed maximum in what the authors confusingly call the yielding strain is consistent with earlier observations, which report on the duration of the transient to reach the steady state, but the effect here is much weaker.

We thank the reviewer for this remark. It seems that in our previous version of the manuscript it was not sufficiently emphasized that all of the results presented in the main text were indeed obtained in this steady state regime. In Extended Data Fig. 1(a) we show the transient process during which the packing fraction reaches the steady state after the particles have been placed randomly in the shear cell. For all Γ , we start the collection of the CT data only if γ_{acc} is larger than 500, i.e., after 1000-5000 shear cycles, depending on Γ . Hence what we present in the main text is definitely not a transient behavior, but the long time dynamics of the system in steady state. This can also be recognized by monitoring the mean squared displacement of the system as a function of the scan sequence, see Fig. R1, which shows that this displacement does not change with the scan sequence, i.e., the system is not aging but indeed in the steady state.

Regarding the yielding point: In a simple shear setup yielding is usually signaled by a quick drop of the stress in a stress-strain curve or by the strain at which the stress starts to become a constant. From the PEL view such a simple shear setup will also give a critical strain $\Gamma \approx 0.1$, i.e., the size of a basin of the PEL. However, in the case of cyclic shear, quite a few particle trajectories are essentially reversible and hence it takes many cycles until the *average* particle has left for good the basin in which it was. Hence it is more useful to use the accumulated strain instead of Γ to parameterize the state of the system, and this is

FIG. 1. Fig. R1: Squared displacement of the system per CT scan as a function of CT scan number after the system has reached the steady state. The absence of a systematic trend is evidence that the system is indeed in a steady state.

what we have done in most of the figures. As a consequence we define the yielding point as the cross-over between the sub-diffusive dynamics (dynamics inside the basin) and the diffusive one (dynamics after the basin is left). This distinction was also the reason why we called the first regime “creep” (signaling a motion that is due to a external driving force but not necessarily out-of-equilibrium) and the second one “flow”. This was seemingly not explained well in the previous version of the manuscript and now we have clarified it in revised manuscript.

As pointed out by the reviewer, we see (Extended Data Fig. 1(c)) that the transient accumulated strain needed to reach the steady state regime shows a mild break at $\Gamma \approx 0.1$, compatible with the singular behavior of $\Delta\gamma_c$ from the steady-state dynamics at this Γ . Although this signature is indeed weaker than reported in previous simulations [e.g., Fiocco et al. PRE **88**, 020301 (2013); Regev et al. Nat. Commun. **6**, 1 (2015)], it is in qualitative agreement with these earlier findings. In view that this transient regime is not the focus of the present work, we do, however, not mention this result in the manuscript.

Modification of the manuscript:

-Added sentence in which we emphasize that the dynamics we present in the main text reflects the steady state dynamics of the system, line 81.

-Clarified the definition of the yielding point, lines 121-122, captions of Fig. 2 and Extended Data Fig 4.

We thank the referees for having brought up these questions since addressing them has helped us to clarify certain points and thus to improve the quality of the manuscript.

REVIEWER COMMENTS

Reviewer #1 (Remarks to the Author):

With pleasure, I have read the revised version of the manuscript "From creep to flow: Granular materials under cyclic shear" by Ye Yuan et al. The significant re-writing has tremendously helped to clear up the story, and present consistent, well-supported results.

I agree with the authors that the difference between the caging behaviour of colloidal/glass forming systems, and the creep behaviour with sub-diffusive dynamics found here is likely related to the absence of thermal motion, and the according exploration of the particle's micro-corrugation in granular systems. The results presented give nice microscopic insight into the internal dynamics of such (micro-corrugated) granular particles. The additional bumpy particle system showing even larger creep regimes and power-laws even closer to diffusion provide a nice confirmation of this hypothesis.

I'm thus happy to recommend publication after clarification of one final point, which I find misleadingly formulated:

On p. 7, the authors write: "For smooth particles one finds caging and diffusion for small and large Γ , respectively". While I agree with the sentence before, I find this sentence unsupported and misleading. The authors have not shown that for smooth granular particles, caging is observed. This is observed for thermal systems, but maybe because of local thermal equilibration rather than smoothness. Or do the authors mean frictionless granular particles as those studied in many simulations? But then, they should give the corresponding reference to support this claim.

When this is clarified, I recommend publication.

Reviewer #2 (Remarks to the Author):

The authors have made some efforts to improve upon the manuscript, but the inclusion of the new data on the BUMP particles, if anything, muddles the picture a bit more. Without going in to the details right away, if the authors' results are to be believed, granular matter made of rough particles will exhibit no solid like, or reversible, regime at all for sufficient cycles of shear. This seems to have very strong consequences to geophysical granular matter, and I suspect there should be relevant literature to compare with in soil mechanics and related areas. It would indeed be worthwhile to make that comparison. In their reply, the authors claim that they prefer to make a distinction between particles with roughness and frictional particles. However, this question has been investigated (see, e.g., PRL 110 (2013) 198002 and subsequent work by O'Hern and co-workers) and as I understand it, the conclusion is that there is no qualitative distinction to be made. And in considering frictional granular matter, there are works (ref. 32 of the authors, and work by Hayakawa and co-workers (Phys. Rev. E 101, 032905, 2020) which indicate that a yielding point can be identified. Thus, there is an inconsistency, which needs to be explained. The authors have taken pains to explain both in the response and the manuscript why they believe that creep-like behaviour will arise because of the landscape of the material they consider. This is not the most important issue, which is why the system does not get

localised, for any finite strain amplitude, asymptotically. A possible explanation is that the system somehow remains marginal, and never compacts beyond the (lowest) jamming point. Can the authors say something with regard to this? In this regard, the results of the final densities reached at different strain amplitudes in extended data fig. 1b is very interesting. I assume that the results are consistent with previous work on granular compaction/dilatancy. Have the authors considered the density effect associated with reaching the diffusive state?

This would be relevant

to having a meaningful interpretation of results. Are the different experiments at different amplitudes started at the same density? (This can easily be done, but I suspect this is not what has been done, nor whether it should be expected to make a qualitative difference).

Can

the experiments be performed at fixed volume conditions? (Again, this will help understand if dilation effects are important). With the same set up, have the authors investigated the behaviour of grains at different depths? The possibility of mobility induced by the surface would be useful. Equally useful would be to reproduce these results numerically. Finally, does the experimental set up permit measurements of stress-strain curves? This will be very revealing.

In summary, I see the results as suggestive of important distinction that potentially exist between granular packings, in the geometry considered, and atomic amorphous solids. However, there are a number of obvious questions that remain unanswered, and there is no satisfactory understanding coming out of the work. I am not convinced about the centrality of the roughness of the particles, though of course it would make a difference. I suspect that the authors are investigating a marginal solid that may well have a vanishing yield strain amplitude, but then the apparent maximum in the crossover strain has no clear meaning. It would be good to ascertain answers to some of these questions to have a clearer statement than what is possible at present.

Reviewer 1 (Remarks to the Author):

With pleasure, I have read the revised version of the manuscript “From creep to flow: Granular materials under cyclic shear” by Ye Yuan et al. The significant re-writing has tremendously helped to clear up the story, and present consistent, well-supported results.

I agree with the authors that the difference between the caging behaviour of colloidal/glass forming systems, and the creep behaviour with sub-diffusive dynamics found here is likely related to the absence of thermal motion, and the according exploration of the particle’s micro-corrugation in granular systems. The results presented give nice microscopic insight into the internal dynamics of such (micro-corrugated) granular particles. The additional bumpy particle system showing even larger creep regimes and power-laws even closer to diffusion provide a nice confirmation of this hypothesis.

I’m thus happy to recommend publication after clarification of one final point, which I find misleadingly formulated: On p. 7, the authors write: ”For smooth particles one finds caging and diffusion for small and large Γ , respectively”. While I agree with the sentence before, I find this sentence unsupported and misleading. The authors have not shown that for smooth granular particles, caging is observed. This is observed for thermal systems, but maybe because of local thermal equilibration rather than smoothness. Or do the authors mean frictionless granular particles as those studied in many simulations? But then, they should give the corresponding reference to support this claim.

We thank the Referee for pointing out this omission. We agree that we have not shown in the manuscript that smooth particles show caging since this result has been documented in various other papers. We now have included the references to these works, Refs. [15–18, 20] (line. 146) to clarify this point.

When this is clarified, I recommend publication.

We thank the referee for this positive assessment and thank him/her for the constructive remarks that have helped us to improve the quality of the manuscript.

Reviewer 2 (Remarks to the Author):

The authors have made some efforts to improve upon the manuscript, but the inclusion of the new data on the BUMP particles, if anything, muddles the picture a bit more. Without going in to the details right away, if the authors' results are to be believed, granular matter made of rough particles will exhibit no solid like, or reversible, regime at all for sufficient cycles of shear. This seems to have very strong consequences to geophysical granular matter, and I suspect there should be relevant literature to compare with in soil mechanics and related areas. It would indeed be worthwhile to make that comparison.

We appreciate that the Referee acknowledges that we have made an effort to improve the manuscript. The Referee is correct in stating that our conclusions are that rough particles under cyclic shear show no solid-like regime. This is one of the main messages of the paper and we agree that this insight is very important for the behavior of geophysical granular matter. We have now added in the Discussion some text and a reference to establish a link with our results and the geoscience applications.

Added text (line 317):

This insight is important in a multitude of situations in geoscience, in which one finds for example that landslide processes are preceded by a complex and slow pre-yield dynamics [7] or small tremors that will affect the long time stability of civil engineering structures such as dams.

In their reply, the authors claim that they prefer to make a distinction between particles with roughness and frictional particles. However, this question has been investigated (see, e.g., PRL 110 (2013) 198002 and subsequent work by O'Hern and co-workers) and as I understand it, the conclusion is that there is no qualitative distinction to be made.

This remark is correct and we are aware of these papers (and cite them in our manuscript). We agree with the Referee that these papers give evidence that particles with friction have similar static properties as particles with roughness. However, to the best of our knowledge, there are no studies which have investigated the influence of roughness on *dynamics*, and also the mentioned papers do not address this question. So one of the main findings of our work is that although the static properties of systems with rough particles are similar to

Fig. R 1. Simulated mean squared displacement of cyclically sheared, frictionless granular disks slightly above the jamming onset. Dimensionless pressure $P \sim \phi - \phi_J$ measures the overjammed level. (a)(b) are captured from Dagois-Bohy *et al.* Soft Matter 13, 9036 (2017), where (a) indicates MSD with various Γ and $P = 10^{-4}$ (green curve refers to $\Gamma = 0.01$), (b) indicates the MSD power exponent as a function of Γ and different colors refer to $P = 10^{-5} - 3 \times 10^{-3}$. (c) is from our preliminary simulations for 1000 bidisperse disks, $P = 4 \times 10^{-4}$, and various Γ (different color).

the ones of particles that have friction, the dynamics is very different. We have now added a sentence in the Discussion emphasizing this insight.

Added text (line 307):

Earlier numerical simulations have given evidence that granular particles with friction have very similar packing properties to system with roughness [36], while...

And in considering frictional granular matter, there are works (ref. 32 of the authors, and work by Hayakawa and co-workers (Phys. Rev. E 101, 032905, 2020) which indicate that a yielding point can be identified. Thus, there is an inconsistency, which needs to be explained. The authors have taken pains to explain both in the response and the manuscript why they believe that creep-like behaviour will arise because of the landscape of the material they consider. This is not the most important issue, which is why the system does not get localised, for any finite strain amplitude, asymptotically. A possible explanation is that the system somehow remains marginal, and never compacts beyond the (lowest) jamming point. Can the authors say something with regard to this?

We thank the Referee for bringing up this point and to remind us of these two papers. We point out that the setup and protocol in these two studies are quite different from ours, and

their main messages also differ from each other. (1) In the 2D experiment of Ref. [32] (now [34]) the authors first prepare a shear jammed state and then probe the associated mechanical/dynamic properties using small cyclic perturbations at a fixed packing fraction. Their main goal is to probe how the preparation history of a shear jammed state, which determines the particle deformation (or the level of overjamming), is connected to the reversibility of the force network. This is irrelevant for our results since we are keeping the system in a mechanically stable state with negligible deformation (i.e., our beads are rigid) and focus on the particle dynamics. (2) In the work of Otsuki and Hayakawa, the authors numerically study the mechanical response of frictional granular systems at various *fixed* packing fractions. It is difficult to compare their results with ours directly since they do not present particle dynamics. Their observed softening (i.e., decay) in the storage modulus can be related to, but is different from, the yielding of particle dynamics, as explained in Ref. [18]. Overall, these are interesting results that contribute to advance our understanding of granular systems in different situations, but not directly relevant to our findings for granular materials formed by rigid particles that maintain mechanically stable under slow external forcing.

Added text (line 82):

Since the beads are very rigid, the obtained packings always maintain mechanical stability under gravity and slow external driving.

Regarding the possibility that our system is exploring a sequence of marginally stable states: This might indeed be a possible scenario since we have rigid particles and shear them in a quasistatic manner. However, the fact that the system is marginal does not make that the MSD at short times (few cycles) is subdiffusive. In fact, the results of Ref. [18] indicate that a marginally jammed system still shows a clear caging regime, see Fig. R1(a) and (b) which are the relevant figures reproduced from that work. We have also carried out some small scale simulations (discrete element method) of hard disks subject to a quasistatic cyclic shear and we find that the MSD does show caging if the strain amplitude Γ is small, see Fig. R1(c). Our findings are also in line with the results by Royer and Chaikin, Ref. [16], who found in their simulations of marginally jammed, frictional granular systems a caging regime for small Γ .

All these mentioned simulations have used models of smooth particles and never shown the pronounced creep dynamics at short times as we discovered. This is thus further evidence that the unique dynamics found in our experiments is related to the roughness of the particles instead of the marginal jamming condition.

Added text (line 141):

It is unlikely that this creep dynamics is related to the motion of the system between marginally jammed states (which can exist for hard particle systems), since previous simulations of systems with such states did show caging for $\Gamma \lesssim 0.1$ [16,18].

In this regard, the results of the final densities reached at different strain amplitudes in extended data fig. 1b is very interesting. I assume that the results are consistent with previous work on granular compaction/dilatancy.

We thank the Referee for this question during the cycling. Our results on the packing fraction ϕ (or density) are indeed compatible with previous work: After the preparation of the initial loose packing, see below for more details on this, the cycling makes that ϕ at strain zero increases with the number of cycles, see Extended Data Fig. 1(a). (Our shear cell, see Fig. 1(a) of the manuscript, does allow to change its volume which means that the packing height will change periodically during cyclic shear.) The value of ϕ in the steady-state, i.e., after a sufficient number of cycles, depends on Γ , see Extended Data Fig. 1(b), and this Γ -dependence is very similar to the simulation results of Otsuki and Hayakawa, PRE **101**, 032905 (2020) (see their Fig.10). In addition, once the steady state has been reached, we can monitor ϕ during one cycle. Figure R2 shows the evolution of this ϕ during a shear cycle (three colors indicate three cycles) for $\Gamma = 0.075, 0.1, \text{ and } 0.125$. One clearly sees that the application of strain leads to a decrease of ϕ , i.e., the system shows dilatancy, and that shear reversal makes that ϕ increases again, i.e., the system shows compaction. Thus this is exactly the behavior expected for a granular under cyclic shear.

Added text (line 302) and Extended Data Fig. 7:

In line with this result the evolution of the packing fraction during a single shear cycle displays persistent hysteresis for different Γ , see Extended Data Fig. 7, similar to the shear

Fig. R 2. (Corresponds to Extended Data Fig. 7) Evolution of the packing fraction during the shear cycle for the ABS system. Data for $\Gamma = 0.075$, 0.125 , and 0.175 are shown, each of covering 2 or 3 cycles after the system has reached the steady state. The persistent hysteresis found here resembles the findings of Ref. [45], indicating the absence of an elastic regime.

stress [45]. This is in contrast to the energy and stress of glasses which show clear reversibility for small [24], thus supporting the view that granular systems and glasses are different.

Extended Data Fig. 7. Evolution of the packing fraction during the shear cycle for the ABS system. Data for $\Gamma = 0.075$, 0.125 , and 0.175 are shown, each of covering 2 or 3 cycles after the system has reached the steady state. The persistent hysteresis found here resembles the findings of Ref. [45], indicating the absence of an elastic regime.

Have the authors considered the density effect associated with reaching the diffusive state? This would be relevant to having a meaningful interpretation of results.

We emphasize that the dynamic behavior that we find at small MSD, i.e., the creep motion, has nothing to do with the transient behavior that occurs during compaction, since all the results presented in the main part of the manuscript are obtained in steady state conditions, i.e., after sufficient number of cycles.

Are the different experiments at different amplitudes started at the same density? (This

can easily be done, but I suspect this is not what has been done, nor whether it should be expected to make a qualitative difference). Can the experiments be performed at fixed volume conditions? (Again, this will help understand if dilation effects are important).

Yes, we have indeed started all the experiments at the same packing fraction ϕ , a random loose packing (RLP). This is documented in Extended Data Fig. 1(a) where the black filled circle shows this initial ϕ . This packing is then cycled with a given strain amplitude Γ and the curves in that graph show how ϕ depends on the total accumulated strain γ_{acc} (and hence on Γ). We note that in fact the value of ϕ for large γ_{acc} is independent of the initial packing fraction since for the measurement of the dynamics presented in the main text we consider only configurations corresponding from the steady state dynamics, i.e., all the transient effects have ceased.

Carrying out the experiments at constant volume: In view of the fact that we are investigating 3D dry granular beads under gravity and our particles are rigid, this is not possible using current setup since (see Fig. R2) the system will dilate.

Added text (line 71 and Extended Data Figure 7, corresponding to Fig. R2):

This design allows to accommodate the granular compaction/dilation during shear.

Extended Data Fig. 7. Evolution of the packing fraction during the shear cycle for the ABS system. Data for $\Gamma = 0.075, 0.125, \text{ and } 0.175$ are shown, each of covering 2 or 3 cycles after the system has reached the steady state. The persistent hysteresis found here resembles the findings of Ref. [45], indicating the absence of an elastic regime.

With the same set up, have the authors investigated the behaviour of grains at different depths? The possibility of mobility induced by the surface would be useful.

This is indeed an important question: As mentioned in the main text, we do all the analyses only after having excluded particles that are located closer than $4d$ (diameter) to the boundary. To check the height dependence of dynamics, Fig. R3 shows the height profile ($h = 0$ means the bottom) of the MSD at a cycle number $\Delta n = 15$ for different Γ . One observes that compared to the clear Γ -dependence, the dynamics changes only mildly with height.

Fig. R 3. (Corresponds to Extended Data Fig. 3) Height profile (in unit of diameter) of MSD at $\Delta n = 15$ for different Γ . h/d is the height from the cell bottom normalized by the small particle diameter, which starts from 4 after excluding boundary particles. One finds very mild height dependence of the dynamics, compared to the clear Γ -dependence.

Added text (l. 93) and Extended Data Fig. 3:
...and height dependence (see Extended Data Fig. 3).

Extended Data Fig. 3. Height profile of MSD at $\Delta n = 15$ for the ABS system. h/d is the height from the cell bottom normalized by the small particle diameter, which starts from 4 after excluding boundary particles. One finds a very mild height dependence of the dynamics, compared to the clear Γ -dependence.

Equally useful would be to reproduce these results numerically.

Yes, we agree that the results presented here could be more comprehensive if supplemented by simulations. However, we expect that standard DEM using frictional hard spherical particles will not generate the creep dynamics that we document in the manuscript. Evidence for this are the results presented in Fig. R1 shown above, as well as the simulation results of Royer and Chaikin, Ref. [16]. Hence a more relevant comparison would be the simulation of *rough* particles and this is a project that we have started. However, such a study is quite involved and thus certainly beyond the scope of the present manuscript.

[Redacted]

Fig. R 4. Evolution of packing fraction (red) and shear force (blue) within one shear cycle for $\Gamma = 0.15$. This figure is reproduced from Ref.[45], PRL **126**, 048002 (2021).

Finally, does the experimental set up permit measurements of stress-strain curves? This will be very revealing.

This is an interesting question. The setup that we have used to carry out the present study does not allow to measure directly the stress-strain curve. However, this curve has been measured in our group in a previous experiment for the ABS beads, and published in PRL **126**, 048002 (2021) (Ref.[45]). The resulting stress-strain curve is reproduced in Fig. R4 for the shear force evolution within one cycle. Although at that time we did not carry out a systematic investigation, we found that the loop formed by force F versus γ displays no qualitative change when changing Γ and in particular we never observed a vanishing hysteresis for small Γ which normally indicates the elastic regime. Accordingly, ϕ versus γ (red curve in Fig. R4 and also the curves in Fig. R2) never becomes a parabola for any Γ , indicating a persistent hysteresis. These two observations support thus the absence of elasticity in our system.

In summary, I see the results as suggestive of important distinction that potentially exist between granular packings, in the geometry considered, and atomic amorphous solids. However, there are a number of obvious questions that remain unanswered, and there is no satisfactory understanding coming out of the work. I am not convinced about the centrality of the roughness of the particles, though of course it would make a difference. I suspect that the authors are investigating a marginal solid that may well have a vanishing yield strain

amplitude, but then the apparent maximum in the crossover strain has no clear meaning. It would be good to ascertain answers to some of these questions to have a clearer statement than what is possible at present.

We hope to have answered the questions raised by the Referee in a satisfactory manner and that in particular have convinced him/her that the phenomena we observe cannot be rationalized by invoking the concept of a marginal solid, since the known marginal solids do not show a creep dynamics. In view that previous studies (mostly numerical) of the dynamics of smooth particle systems (with or without friction) have always reached the conclusion that there is caging, it seems to us that the most plausible explanation for the observed creep dynamics is related to the surface roughness of the particles. As the Referee points out, this insight is important not only for fundamental science but also for geophysical domains, and we do find relevant studies investigating pre-yield slow dynamics caused by complex tiny perturbations in real landslide processes. Therefore we think that this manuscript does merit to be published in a journal like Nature Communications.

REVIEWER COMMENTS

Reviewer #1 (Remarks to the Author):

The authors have adequately addressed my remaining point, and I'm happy to recommend publication.

Reviewer #2 (Remarks to the Author):

I have read the revised manuscript and the responses of the authors. The authors have made an effort to address comments, though not always convincingly. For the systems they consider, the results indicate that there is always a transition from creep to diffusive motion, and the authors contrast their results with glasses, which have been seen to exhibit a transition from a pre-yield regime where they become non-diffusive to a post-yield diffusive regime. There are several comparisons to make in arriving at an understanding of what the authors are seeing, which includes comparison with glasses, typically studied at constant volume conditions, soft sphere models of granular matter, frictional granular matter, etc, studied under cyclic deformation. In their responses to some of the questions raised, the authors have argued that the various cases of granular systems/models exhibit differences of various kinds from their system and protocol, and thus the fact that caging or non-diffusive states exist in these cases does not have implications for their results, and as part of that logic, they also arrive at the conclusion that the peculiar behaviour they observe can be traced to the roughness of the particles they investigate, which should be distinguished from frictional particles. This logic has some degree of plausibility, but certainly does not help explain the authors' results in the context of the other results in an insightful manner. It also raises questions about how general one should view the authors results to be, and therefore, how significant.

I provide some specific points below.

In response to the suggestion to compare with the work of O'Hern et al, who treated rough particles and compared them with frictional particles, and concluded that the two cases compare, the authors state "However, to the best of our knowledge, there are no studies which have investigated the influence of roughness on dynamics." But this is not correct. The PRL paper mentioned discussed the vibrational density of states, and subsequent papers have studies dynamical effects such as sedimentation etc. A more convincing comparison needed to have been attempted.

In response to the suggestion to compare with the work of Zhao et al (PRX 2022) and Hayakawa and coworkers, the authors stated that these other studies used different protocols and therefore they could not be compared. However, these are granular systems which exhibit stability under cyclic shear, and therefore are centrally important to comment upon.

In discussing other studies (ref. 18 and their own simulations), the authors conclude that the fact that these studies exhibit caging, but their results do not as evidence that the roughness of the particles is the key feature of importance. There are several differences in these studies, which should be more clearly analysed. A particular difference is the open boundaries in the experiments.

This is nevertheless a shared feature with the frictional simulations of Royer and Chaikin (ref. 16) who do see non-diffusive states. The authors see this as an argument in favour of their hypothesis that the roughness of their grains is the key feature. This may be acceptable if the authors had other ways of arguing that frictional and rough grains should indeed be treated as being distinct. The authors have not presented such a case. Nevertheless, given that the simulations share significant features of the authors' experiments, the comparison may well support the authors' claim.

I feel that they may indeed be wrong in their conclusion (now added as a comment) that marginality is unlikely to be important. The specific details of the authors' protocol may well mean that they are able to probe only marginally stable packings. Addressing this question may require additional investigation, but I am not convinced that there is enough understanding to exclude this possibility.

Additional remark: The new Extended Data Fig. 7 is very interesting, but the strain amplitudes are indeed quite high. Do the authors have a data set for smaller values, say 1%? They may or may not see hysteresis in that case.

In my previous report, I suggested that the authors consider the implications of their work to geophysical contexts since the implications would be significant. The authors have now added a comment to that effect, but it would have been desirable to consider available evidence in that context and compare notes. I would like to suggest in particular the interesting work of Jerolmack et al (Nature Comm 2021 12:3909) that should be consulted and commented upon.

Given these considerations, one has to treat the manuscript as one that reports results that, if they are right, are surprising and possibly of broad interest, but for which there isn't yet a satisfactory understanding. I strongly urge, however, that the authors revisit these points and address them more satisfactorily.

Reviewer #3 (Remarks to the Author):

I was asked to review this manuscript after it had received contrasting reviews and a revision.

I agreed to review the paper in light of this, but in the meantime I had a conference and then got COVID, so I am in much diminished capacity. Nonetheless, I did read the paper and the reviews; so rather than submit nothing at this late stage, I thought I would share a quick response.

The authors have a thorough and careful study of the response of granular media to cyclic shear. In particular, the explicit examination of particle roughness, and the 3D imaging of grains, makes this study special.

That said, this study does build on significant previous work by the authors. The focus on particle roughness is the new contribution.

The most important result, it seems, is that the mean square displacement (MSD) does not show caging at short times for grains -- as it does for glass simulations and experiments of soft solids. The authors take this to mean that the granular material has no elastic regime. It seems that both reviewers acknowledge this point as well.

I was suprised to not see literal trajectories shown in the main text. Since elasticity means that particles follow the same path in cyclic strain, it seems like they should show that the path is not identical. And is there anything interesting, like loops, as seen in the work of Arratia's experiments (e.g., Galloway, Arratia and Jerolmack, *Soft Matter* 2020)?

Since the main point is that granular material, or at least rough granular material, has no elastic regime, it would be really nice to have a rheologic demonstration of this. Review 2 brings up stress-strain curves, which would do the trick. It seems that the current setup can't do rheometry, which is fair...but a macroscopic demonstration of the mechanical behaviors inferred from the microscopic would prove the point.

To be honest, a lot of the text is written and targeted at the specialist. This is fine, but I had trouble following some of the nuanced arguments -- especially because the authors tended to define their own parameters for things like structure and disorder rather than use some widely used ones (for example: dynamical heterogeneity).

From the perspective of the two reviews, however, it seems the authors have gone as far as they can with the setup that they have. I might like to see the manuscript edited to be a bit more accessible (I am reasonably well versed in soft matter and granular physics, and still found myself saying "Wait, zoom out, what are we talking about here?" many times in the manuscript). But I think the paper is worth publishing because the experimental results are strong and will lead to new insights and work by others.

Reply

Reviewer 1 (Remarks to the Author):

The authors have adequately addressed my remaining point, and I'm happy to recommend publication.

We are happy to read that Reviewer 1 is satisfied with the text and recommends publication. Furthermore we thank this person for having commented in a constructive manner the various points raised in the reports by Reviewer 2. This has allowed us to address these points in an efficient manner.

Reviewer 2 (Remarks to the Author):

I have read the revised manuscript and the responses of the authors. The authors have made an effort to address comments, though not always convincingly. For the systems they consider, the results indicate that there is always a transition from creep to diffusive motion, and the authors contrast their results with glasses, which have been seen to exhibit a transition from a pre-yield regime where they become non-diffusive to a post-yield diffusive regime. There are several comparisons to make in arriving at an understanding of what the authors are seeing, which includes comparison with glasses, typically studied at constant volume conditions, soft sphere models of granular matter, frictional granular matter, etc, studied under cyclic deformation. In their responses to some of the questions raised, the authors have argued that the various cases of granular systems/models exhibit differences of various kinds from their system and protocol, and thus the fact that caging or non-diffusive states exist in these cases does not have implications for their results, and as part of that logic, they also arrive at the conclusion that the peculiar behaviour they observe can be traced to the roughness of the particles they investigate, which should be distinguished from frictional particles. This logic has some degree of plausibility, but certainly does not help explain the authors' results in the context of the other results in an insightful manner. It also raises questions about how general one should view the authors results to be, and therefore, how significant.

Comment of Reviewer 1: I have carefully read the previous revision round, including

critical points of Reviewer 2 and answers of the authors, and the current comments of Reviewer 2. My point of view is the following: To contextualize the referee comments, I remind that the current paper is experimental in nature, and it achieves, using an elaborate newly applied experimental technique (x-ray tomography), unique new insight into the dynamics of a sheared granular system. As such I find the paper valuable and sufficiently novel, providing new experimental access to 3D microscopic granular dynamics, worth publishing in Nature Comm. Reviewer 2 argues mostly with simulation studies and their findings. While these are of course very valuable on their own, I believe that in the end, these remain simulations, while experiments have their own rights and own independent value. So in my opinion, these simulation studies do not compromise the novelty of the present carefully conducted experiments. However, I do agree with the reviewer that to further increase the impact of the current work and possibly advance its understanding, the authors could include some of the discussions/comparisons with these other works suggested by the referee. I provide some more detailed perspectives on the referee's comments below (in red).

Reply: Reviewer 2 acknowledges that we have made an effort to address the previous criticism and that the arguments that we put forward to arrive to the conclusion that roughness is an essential ingredient to rationalize our findings are “plausible”, and we are pleased with this judgement. On the other hand, the Reviewer criticizes that we do not rationalize our findings by means of the results of previous studies. We point out that these previous studies were to a large extent numerical simulations, while we are dealing with real experimental systems. (This fact was also pointed out by Reviewer 1 who argued correctly that both techniques have their own right of existence.) In addition we insist that the goal of simulations is to investigate the properties of particle systems in which one leaves out ingredients that are considered irrelevant for the phenomenon that one wants to study, since this allows to get a clearer picture. As a consequence many simulations of granular materials have been done for model systems that are idealized: One uses ideal hard spheres, no polydispersity, no friction, no roughness, periodic boundary conditions, volume perfectly constant, etc. Despite these approximations, such simulations have turned out to be most useful since they allow to probe properties of the system that a real experiment cannot access. However, one cannot expect that simulations are able to reproduce *all* properties of the real systems since some relevant ingredient might have been left out. Deciding *a priori* whether or not a simulation indeed includes all relevant ingredients is impossible and hence

one finds this out only by comparing the obtained results with the ones from experiments. So the message of our manuscript is not that the previous simulations are wrong, but that there might be a property of granular systems, roughness, that has not been taken into consideration in a sufficient manner for understanding the relaxation dynamics of these systems. We have reached this conclusion by applying Occam’s razor, but we certainly cannot exclude the possibility that a more complex mechanism is at work. So the presented results should be taken as an invitation to probe in the future this question in more detail. Having said this, we agree with Reviewer 1 that it is important to put forward the apparent (?) contradictions between our results and the simulation results in a more direct manner. Similar points have been raised below and hence we reproduce the text that we have added to the manuscript (lines 151-158) later.

I provide some specific points below.

In response to the suggestion to compare with the work of O’Hern et al, who treated rough particles and compared them with frictional particles, and concluded that the two cases compare, the authors state “However, to the best of our knowledge, there are no studies which have investigated the influence of roughness on dynamics.” But this is not correct. The PRL paper mentioned discussed the vibrational density of states, and subsequent papers have studies dynamical effects such as sedimentation etc. A more convincing comparison needed to have been attempted.

Comment of Reviewer 1: The papers by O’Hern concern simulations; the present work is experimental, and as such provides a valuable complementary view on the microscopic yielding dynamics of granular suspensions. In this sense, I agree with the authors that there are no studies (experimental studies) investigating the influence of roughness on the microscopic dynamics. To make this clear, they could add the word “experimental”, i.e. “there are no experimental studies which have investigated the influence of roughness on dynamics”

Reply: We thank Reviewer 2 for reminding us that in the mentioned work by O’Hern the dynamical matrix was studied. However, when we referred to “dynamics”, we did not have in mind the vibrational dynamics, but the relaxation dynamics. We have now corrected this misunderstanding in the text, and, following the suggestion of Reviewer 1, we emphasize that we refer to experimental work. In addition, we have looked at the subsequent paper

by the O’Hern group [Clark et al. PRF **2**, 034305 (2017)], evoked by Reviewer 2, in which the authors modelled the dynamics of granular particle sediments in a liquid environment. That study aimed to gain understanding on the subtle flowing transition occurring when such a system is sheared. We point out that the presence of viscous damping forces and the velocity profile along gravitational direction make their dynamics very different from the quasistatic motion in our experimental system. In addition, although some results of that study were for non-circular particles, these asperities consisted only of 5 (!) “bumps” around a central particle, i.e., such particles are rather different from the rough particles considered in our study. In view of these differences we do not include a comparison/discussion in the manuscript.

Modification of sentence (line 60): Also the connection between surface roughness and friction of granular particles has been investigated only recently [Papanikolaou et al. PRL **110** 198002 (2013); Ikeda et al. PRL **124** 208001 (2020); Sun et al. PRL **125** 268005 (2020)], but their influence on the particle-level relaxation dynamics remains unknown, especially for experiments.

In response to the suggestion to compare with the work of Zhao et al (PRX 2022) and Hayakawa and coworkers, the authors stated that these other studies used different protocols and therefore they could not be compared. However, these are granular systems which exhibit stability under cyclic shear, and therefore are centrally important to comment upon.

Comment of Reviewer 1: Hayakawa and coworkers is also simulation work, while Zhao et al (PRX 2022) is experimental (photoelastic discs visualizing the force network). Indeed I find both reasonably relevant to the current paper, so I agree that a short discussion of the works in the light of the findings of the current paper would be worthwhile and would add impact to the current manuscript.

We thank both Reviewers for this comment. We have now included in the text a discussion of these studies. In our reply to the next point (see below), we will in addition comment in detail on these papers.

Added text:

(1) Line 56: The inconsistency between these studies as well as the complex dependence on the shear protocol [Otsuki & Hayakawa PRE **101**, 032905 (2020); Zhao et al. PRX **12**, 031021 (2022)] have not been elucidated.

(2) Lines 151-158: Furthermore, we recall that the dynamics of granular systems depends significantly on the details of the driving protocol. For example, simulations using a fixed-volume condition have given evidence that jammed and unjammed states can coexist, resulting in a complex response as a function of both packing fraction and Γ [Das et al. PNAS **117**, 10203 (2020); Otsuki & Hayakawa PRE **101**, 032905 (2020)]. In addition, it was found that the preparation history of a jammed state in an experimental soft particle systems can alter the range of Γ in which the dynamics is reversible [Zhao et al. PRX **12**, 031021 (2022)]. However, since our system corresponds to the hard-particle limit at a constant pressure, we expect that these earlier findings are not relevant for the interpretation of our results.

In discussing other studies (ref. 18 and their own simulations), the authors conclude that the fact that these studies exhibit caging, but their results do not as evidence that the roughness of the particles is the key feature of importance. There are several differences in these studies, which should be more clearly analysed. A particular difference is the open boundaries in the experiments.

Comment of Reviewer 1: I agree with the referee that the authors should be careful to not overclaim their results, i.e. to conclude strictly along their observations on the effect of roughness of the particles. However, within their experimental observations of the two granular systems studied, I believe firm conclusions can be drawn.

Reply: We agree with both Reviewers that our experimental results are not a strict proof that roughness is the only important factor to rationalize our findings. In order to avoid a repetition of the arguments, we will present our logic in the reply to the next point and reproduce there also the text we have added to the manuscript.

This is nevertheless a shared feature with the frictional simulations of Royer and Chaikin (ref. 16) who do see non-diffusive states. The authors see this as an argument in favour of their hypothesis that the roughness of their grains is the key feature. This may be acceptable if the authors had other ways of arguing that frictional and rough grains should indeed be treated as being distinct. The authors have not presented such a case. Nevertheless, given that the simulations share significant features of the authors' experiments, the comparison may well support the authors' claim. I feel that they may indeed be wrong in their conclusion

(now added as a comment) that marginality is unlikely to be important. The specific details of the authors' protocol may well mean that they are able to probe only marginally stable packings. Addressing this question may require additional investigation, but I am not convinced that there is enough understanding to exclude this possibility.

Comment of Reviewer 1: Indeed this will be difficult to pinpoint, and in my opinion is also beyond the scope of the present manuscript, so I don't think this should be over-speculated.

Reply: We thank Reviewer 2 for raising these points. After a careful inspection of the mentioned references, we present in the following a detailed discussion which we hope will clarify things. In brief: (1) For highly compressed granular systems under cyclic shear, reversibility for small shear strain Γ is caused by a stable contact network. This reversibility is, however, not identical to (nor necessary for) caging. Different from such compressed system above the jamming point, marginally jammed granular system under cyclic shear can display caged dynamics (as revealed by previous simulations) regardless of the possible rearrangement in their contact network. (2) Different from previous simulations, our experiment shows for none of the investigated Γ any sign of caging (which is nontrivial), and hence we rationalize this finding by the presence of particle roughness. This is of course not a conclusion that one is forced to make. However, we support our point of view by investigating the qualitative and quantitative dependence of the relaxation dynamics on the roughness and find that this parameter does indeed strongly influence the relaxation dynamics. (3) The dynamics of granular systems shows a complex dependence on the protocol, especially close to the jamming point. Certain mechanically stable states, reported in previous studies [Zhao et al. PRX **12**, 031021 (2022); Otsuki & Hayakawa PRE **101**, 032905 (2020)], exist only under constant-volume condition, i.e., the particles need to be deformable. These states can give rise to a reversible dynamics (caused by a finite distance above jamming onset), but this should not be confused with the caging dynamics. Our hard-particle system cannot be probed under constant volume conditions and hence this type of reversible dynamics cannot be accessed. Thus these previous studies do not contradict our findings.

In the following we give more details on the listed points.

(1) Previous studies of granular dynamics under cyclic shear: From marginal jamming to strong over-jamming:

A key parameter for a jammed granular system is the dimensionless pressure P (stress

level normalized by particle stiffness), which determines the overlap between the particles and thus the excess density with respect to the jamming point. For large P , i.e., a highly compressed/over-jammed soft-sphere/granular system, the dynamics under cyclic shear is similar to the one of thermal model glasses, i.e., one observes clear reversibility for small shear amplitude Γ , i.e., $\Gamma \lesssim 0.1$, and irreversibly/diffusion for large Γ , see for example [Kawasaki & Berthier PRE **94**, 022615 (2016); Das et al. PNAS **117**, 10203 (2020)]. For the case $P \rightarrow 0$, one expects caging for a small but finite Γ . This is seen in Figs. R1(a)(b) from Dagois-Bohy et al. Soft Matter **13**, 9036 (2017) which shows a MSD that indicates caging if Γ is not too large and $\Gamma \gg P$. This behavior is also found in simulations of frictional granular materials with $P \rightarrow 0$, see [Royer & Chaikin PNAS, **112**, 49 (2015)]. So for the case $P \rightarrow 0$ the dynamics is caged even if most individual contacts are broken by a small but finite strain (the strain limit to maintain the same contact network is $\sim P$). In other words, caging does not request the presence of persistent contacts. This situation is thus similar to the case of hard-sphere glasses for which one has caging without a stable contact network.

Between the mentioned two limiting cases a dynamic phase diagram emerges that clarifies the role of jamming marginality, see Fig. R1(c) reproduced from the simulation study by Dagois-Bohy et al. Soft Matter **13**, 9036 (2017). In this figure, the “yielded” regime corresponds to the diffusive dynamics, and the “linear” regime corresponds to reversibility since one finds that for small P and $\Gamma < P$ the contact network hardly changes. For small P and $P < \Gamma \lesssim 0.1$, a “softened” regime emerges which corresponds to caged dynamics, in which the power-law exponent of the MSD ($=\alpha$) is small, Fig. R1(b), for $\Gamma \lesssim 0.1$ and very small P . Thus the important point is to distinguish at these small P ’s the regime with strict reversibility (which is related to jamming marginality) from the caging regime. For large P , the softened regime disappears, see Fig. R1(d), and the two transitions merge at $\Gamma \approx 0.1$, which agrees with Das et al. PNAS **117**, 10203 (2020)] for soft spheres under cyclic shear at fixed high packing fractions.

(2) The absence of caging in our experiment:

In our system, the hard beads are stabilized by gravity (thus weak pressure), leading to $P \rightarrow 0$ and $\Gamma \gg P$ and hence the strict reversibility becomes irrelevant and also inaccessible. For small Γ , the softened/caged regime seen in Fig. R1(c) is observed in the simulation [Royer

Fig. R 1. (a)-(c): Reproduced from Dagois-Bohy et al. *Soft Matter* **13**, 9036 (2017), Fig.2(b), Fig.3(b), and Fig.1(e). γ_0 is the cyclic shear amplitude (our Γ) and P is the dimensionless pressure. (a): MSD with various Γ and $P = 10^{-4}$ (curves from green to blue correspond to $\Gamma = 0.01 - 1$). (b): MSD power-law exponent as a function of Γ and different colors refer to $P = 10^{-5} - 3 \times 10^{-3}$. (c): Dynamic phase diagram of P and Γ . Yielded: Diffusive regime; Linear: Reversible regime; Softened: Caging regime. For small P , the reversible regime is related to the jamming marginality and its upper strain boundary is $\sim P$, different from the yielding strain scale. For large P , one has only a single transition between the reversible regime and the yielding. (d): Reproduced from Das et al. *PNAS*, **117**, 10203 (2020), Fig. 6(c). For very high packing fraction beyond jamming, the dynamics is either reversible or diffusive/irreversible, in agreement with the trend that two transitions merge at large P in panel (c).

& Chaikin *PNAS*, **112**, 49 (2015)] but is absent in our experiments. Instead, we observe a persistent sub-diffusive regime at small strain scale for all Γ and both systems. We speculate that this difference is due to the roughness of the particles. As pointed out by Reviewer 2, we have not presented a systematic study how the static and dynamic properties of frictional grains differ from the ones of rough grains. Such a study is well beyond the scope of the

research presented here. Hence we agree with both Reviewers that we cannot present a hard proof that roughness is the only important factor, and we certainly do not want to make such a claim. In an experimental system, it is impossible to turn off completely roughness or friction and as a consequence there will always be the possibility that the interpretation of the results is flawed. However we have made significant efforts to substantiate our idea by changing the roughness of the particles and exploring its influence on the relaxation dynamics. We find that the dynamic crossover in Γ for the rougher system (BUMP) almost disappears, which we rationalize by its more rugged landscape in comparison with the smooth one of the ABS system.

(3) Comments on several other references:

Regarding the experimental work of [Zhao et al. PRX **12**, 031021 (2022)]: The pre-shear treatment of the granular system at constant volume induces a non-negligible finite P due to the dilatancy. So in the subsequent cyclic shear, if Γ is so small that the shear cannot overcome this P , the system shows reversibility (or it is “ultrastable” as the authors put it). This is the reason why the phase boundary between jammed and ultrastable regime strongly depends on the initial strain, since the latter obviously determines P , see Fig. R2. Note that in this figure one has $\Gamma \ll 0.1$, which is another evidence that in that work the authors explore a transition that is different from ours. Moreover, these authors found that only a unjammed regime exists apart from the “ultrastable” regime, which is different from the predicted softened regime indicated in Fig. R1(c). We speculate that this difference is related to the fact that they considered the case of constant volume instead of the constant pressure used in Fig. R1(c) and our experiment.

The paper by Otsuki & Hayakawa PRE **101**, 032905 (2020): In that work the authors look at the rheological aspect of the same problem, again under constant-volume condition. To understand these results it is helpful to recall first a result by Dagois-Bohy et al. Soft Matter **13**, 9036 (2017), reproduced in Fig. R3(a). For small P the storage modulus vs. Γ (their γ_0) shows a two-step behavior (softening and yielding) but only a one-step decay for larger P , in agreement with Fig. R1(c). The plateau before the softening corresponds to the reversible regime, in which the contact network remains stable for small Γ . Rescaling γ_0 by P , Fig. R3(b), indicates that the reversible regime is observed for $\Gamma \lesssim P$ for small P while for large P this regime is no longer observed, which agrees with Fig. R1(c). Differently, Fig. R3(c), reproduced from the work of Otsuki & Hayakawa, shows the complex behavior of the

[Redacted]

Fig. R 2. Reproduced from Zhao et al. PRX **12**, 031021 (2022), Fig. 2(d). The initial strain shear-jams a configuration with a finite P , and hence controls the phase boundary. Here $\delta\gamma$ indicates our Γ .

storage modulus, G' , for various fixed packing fractions. One recognizes that if ϕ is large, G' decays at $\Gamma \approx 10^{-2} - 10^{-1}$, which implies a single reversible-irreversible, i.e., diffusion, transition for large P given in Fig. R1(c). The curves for smaller ϕ , i.e., close to the jamming point, are not that easy to interpret. The reason is that the jamming-unjamming transition is probed at constant-volume, which gives rise to a complex phase diagram, see Fig. R3(d). A similar phase diagram has also been reported in Das et al. PNAS, **117**, 10203 (2020) obtained from simulations of a soft sphere system.

Our experiment is done at constant-pressure condition, thus allowing for volume change due to compaction/dilation, and $P \rightarrow 0$ and $\Gamma \gg P$. Hence the P -dependent reversible regime, responsible for the mechanical stability in the above two studies, is irrelevant for our system. Also our setup cannot access the unjammed states, since gravity makes that our system is always in the jammed state.

(4) Scenario of hard-sphere marginal glasses:

In addition to the above-mentioned case for marginal jamming in which the jamming point is approached from high densities, studies on hard-sphere glasses [Jin et al. Science Advances **4**, eaat6387 (2018)] have revealed a complex phase diagram as a function of ϕ and Γ for unjammed states. In this case, a so-called marginal glass phase can be distinguished

[Redacted]

Fig. R 3. (a) and (b): Storage modulus from a cyclic shear simulation as a function of shear amplitude (γ_0 stands for our Γ) for different P (dark to light curves indicate increasing P from 10^{-6} to 0.1), reproduced from Dagois-Bohy et al. *Soft Matter* **13**, 9036 (2017), Figs. 1(a)(c). (c): Similar plot as panel (a) but for different packing fractions ϕ ($\gamma_0^{(I)}$ stands for our Γ). (d): Jamming phase diagram as a function of ϕ and Γ . Panels (c) and (d) are reproduced from Otsuki & Hayakawa *PRE* **101**, 032905 (2020), Figs. 6 and 8, respectively.

from a normal hard-sphere glass for a dense packing below jamming. Interestingly this glass marginality is different from the jamming marginality (originated from mechanical equilibrium): The hallmark of such a marginal glass is a logarithmic growth of the mean-squared displacement [Berthier et al. *PNAS*, **113**, 8397 (2016)], instead of the pronounced plateau for a normal glass. Since we are working on mechanically stable granular systems displaying persistent sub-diffusion instead of a plateau, we do not consider the case of glass marginality when rationalizing our findings.

Added text:

(1) Paragraph change (Lines 139-147): Depending on whether the jammed system is close to

the hard-particle limit [Mailman et al. PRL **112**, 228001 (2014); Royer & Chaikin PNAS, **112**, 49 (2015); Dagois-Bohy et al. Soft Matter **13**, 9036 (2017)] or highly compressed [Kawasaki & Berthier PRE **94**, 022615 (2016); Dagois-Bohy et al. Soft Matter **13**, 9036 (2017); Das et al. PNAS, **117**, 10203 (2020)], caging at small Γ can originate from either the confining effect due to the nearest neighbors, similar to the case of a dense hard-sphere packing, or the persistent contact network. The latter case, sometimes specified as reversibility [Das et al. PNAS, **117**, 10203 (2020)], is however not necessary for caging. In contrast to these simulations, we find a persistent sub-diffusive dynamics for all Γ at small $\Delta\gamma$. Our results are also different from a hard-sphere marginal glass which does show a logarithmic growth in MSD [Jin et al. Science Advances **4**, eaat6387 (2018); Berthier et al. PNAS, **113**, 8397 (2016)]. Thus we conclude that the presence of roughness alters radically the dynamics of granular systems, i.e., sub-diffusion is always present.

(2) Paragraph change (Lines 312-319): However, earlier simulations of smooth particles, e.g., [Royer & Chaikin PNAS, **112**, 49 (2015)], did not observe the sub-diffusive dynamics. Therefore we argue that this type of dynamics is due to the PEL's micro-corrugation which originates from the surface roughness of particles, intrinsic for real granular materials. A comparison between the ABS and BUMP systems demonstrates the significant influence of roughness on the dynamics as well as the underlying PEL. Further studies, in which roughness is systematically varied, will be helpful to elucidate the details of the observed creep dynamics.

Additional remark: The new Extended Data Fig. 7 is very interesting, but the strain amplitudes are indeed quite high. Do the authors have a data set for smaller values, say 1%? They may or may not see hysteresis in that case.

Reply: We point out that the hinges of our shear box have a mechanical precision that is finite. This limits the lower resolution of Γ , and hence strain amplitudes that are too small cannot be realized. However, we can determine ϕ vs. γ curves for $\Gamma = 0.03$ and include them in the updated Extended Data Fig. 7. For the sake of a better visualization, we use a solid curve for the forward direction of shear (increasing γ) and a dashed curve for the backward direction. (To give an idea on the accuracy of the data we show two consecutive cycles.) One clearly observes a similar hysteresis for all four Γ . So this is evidence that in the system there is indeed no elastic regime for these values of Γ .

Fig. R 4. Updated Extended Data Fig. 7. Evolution of the packing fraction during the shear cycle for four different Γ . In each panel we show the evolution during two cycles.

Added text: Updated Extended Data Fig. 7. with caption.

In my previous report, I suggested that the authors consider the implications of their work to geophysical contexts since the implications would be significant. The authors have now added a comment to that effect, but it would have been desirable to consider available evidence in that context and compare notes. I would like to suggest in particular the interesting work of Jerolmack et al (Nature Comm 2021 12:3909) that should be consulted and commented upon.

Comment of Reviewer 1: I agree with this comment, worth looking at and possibly briefly discussing the suggested reference; it may give the paper a broader appeal.

We thank Reviewer 2 for suggesting this interesting and relevant work Deshpande et al. Nat. Comm. **12**, 3909 (2021). We have now added some text in the Discussion on it.

Added text (Lines 326-331): Interestingly, a recent experiment reported the persistent creep of a sandpile even without an apparent external disturbance [Deshpande et al. NC (2021)], which indicates that the particle-level relaxation can be triggered by multiple microscopic processes. This finding is thus in line with our results, in that we reveal the absence of caging even if Γ is very small. Hence granular materials have no well-defined stability threshold, in stark contrast to other amorphous solids.

Given these considerations, one has to treat the manuscript as one that reports results that, if they are right, are surprising and possibly of broad interest, but for which there isn't yet a satisfactory understanding. I strongly urge, however, that the authors revisit these points and address them more satisfactorily.

Although we are highly confident that our results do reveal that particle roughness is a very important parameter for understanding the relaxation dynamics of real granular systems, we are at this point not able to prove this in a completely unambiguous manner because current experimental limitations prevent to do the necessary studies. Therefore one could indeed argue like the Reviewer that the understanding is not "satisfactory". However, we think that such a verdict is not quite just, since we do provide in the manuscript an explanation that, as the Reviewer acknowledges, is "plausible". So we think that the present work is indeed an important step forward in the field of granular materials since it is likely to trigger further studies that probe the relevance of roughness for these systems. To emphasize this point we have added the following text:

Added text (Lines 317-318): Further studies, in which roughness is systematically varied, will be helpful to elucidate the details of the observed creep dynamics.

Reviewer 3 (Remarks to the Author):

I was asked to review this manuscript after it had received contrasting reviews and a revision.

I agreed to review the paper in light of this, but in the meantime I had a conference and then got COVID, so I am in much diminished capacity. Nonetheless, I did read the paper and the reviews; so rather than submit nothing at this late stage, I thought I would share a quick response.

The authors have a thorough and careful study of the response of granular media to cyclic shear. In particular, the explicit examination of particle roughness, and the 3D imaging of grains, makes this study special. That said, this study does build on significant previous work by the authors. The focus on particle roughness is the new contribution.

Reply: We appreciate that the Reviewer acknowledges the novelty of this work and regards it as “thorough and careful”.

The most important result, it seems, is that the mean square displacement (MSD) does not show caging at short times for grains – as it does for glass simulations and experiments of soft solids. The authors take this to mean that the granular material has no elastic regime. It seems that both reviewers acknowledge this point as well. I was surprised to not see literal trajectories shown in the main text. Since elasticity means that particles follow the same path in cyclic strain, it seems like they should show that the path is not identical. And is there anything interesting, like loops, as seen in the work of Arratia’s experiments (e.g., Galloway, Arratia and Jerolmack, *Soft Matter* 2020)?

Reply: Yes, it seems that all Reviewers agree that there is no caging regime in our system. We thank this Reviewer for the suggestion to look at the trajectories of the particles during a single cycle and see whether they form loops. Unfortunately we do not have the CT scans that are needed to get these trajectories. However, the value of the MSD after one cycle is finite which demonstrates that at least some of the particles are moving. The Van Hove function, Fig. 3 of the manuscript, shows that there is no δ -peak at $d_x = 0$, i.e., the fraction of particles that are *not* moving (because their trajectory forms a loop) must be small. Thus

this is good evidence that there is no clear distinction between immobile and mobile particles (one usually expects this in a standard glass) in our system, and typical particle motion is a slow but steady exploration of configuration space. Hence reversible motion is irrelevant. This leads to the conclusion that the system does not have an elastic regime, at least not for the values of Γ that we can probe. We have now added some text to include this point when discussing the Van Hove function.

Added text (line 167): ... and there is no clear distinction between immobile and mobile particles. So the reversible particle motion is irrelevant, which is consistent with the absence of an elastic regime as mentioned above.

Since the main point is that granular material, or at least rough granular material, has no elastic regime, it would be really nice to have a rheologic demonstration of this. Review 2 brings up stress-strain curves, which would do the trick. It seems that the current setup can't do rheometry, which is fair...but a macroscopic demonstration of the mechanical behaviors inferred from the microscopic would prove the point.

We agree with the reviewer that having rheological data would indeed be very useful. Unfortunately the setup we used to get our data does not allow to measure the stress-strain curve. However, earlier experiments on a similar system have measured the stress-strain curve and found that there is indeed no elastic regime [PRL **126**, 048002 (2021)], see Fig. R5. The associated ϕ vs. γ curve also has two branches, see figure. This latter curve is qualitatively very similar to the ones we have measured in our setup, see Fig. R4. Therefore we expect that in our system we do indeed have no elastic regime, at least not for strain amplitudes $\Gamma \geq 0.03$.

To be honest, a lot of the text is written and targeted at the specialist. This is fine, but I had trouble following some of the nuanced arguments – especially because the authors tended to define their own parameters for things like structure and disorder rather than use some widely used ones (for example: dynamical heterogeneity).

We thank the reviewer for this remark. As a consequence we have carefully re-read our manuscript and to adapt at multiple places our wording and the nomenclature to the one used in the previous literature.

[Redacted]

Fig. R 5. Evolution of packing fraction (red) and shear force (blue) within one shear cycle for $\Gamma = 0.15$. The system is the ABS bead packing. Note that the force $F(\gamma)$ has hysteresis, indicating that the system is not elastic. This figure is reproduced from PRL **126**, 048002 (2021).

From the perspective of the two reviews, however, it seems the authors have gone as far as they can with the setup that they have. I might like to see the manuscript edited to be a bit more accessible (I am reasonably well versed in soft matter and granular physics, and still found myself saying "Wait, zoom out, what are we talking about here?" many times in the manuscript). But I think the paper is worth publishing because the experimental results are strong and will lead to new insights and work by others.

We thank the reviewer for this remark. As a consequence we have made an effort to re-write parts of the manuscript in an attempt to make it more accessible. Finally we thank this referee for his/her various constructive comments that have helped us to improve the quality of the manuscript.

REVIEWERS' COMMENTS

Reviewer #1 (Remarks to the Author):

The authors did a very thorough job to respond adequately to all reviewer comments. They have well argued and adjusted the manuscript to implement more literature comparison and discussion, putting this work in a good context. I therefore recommend publication.